# Centromeres are maintained by fastening CENP-A to DNA and directing an arginine anchor-dependent nucleosome transition

Lucie Y. Guo[1,2], Praveen Kumar Allu[1], Levani Zandarashvili[1], Kara L. McKinley[3], Nikolina Sekulic[1,†], Jennine M. Dawicki-McKenna[1], Daniele Fachinetti[4,†], Glennis A. Logsdon[1,2], Ryan M. Jamiolkowski[5], Don W. Cleveland[4], Iain M. Cheeseman[3] & Ben E. Black[1,2]

Maintaining centromere identity relies upon the persistence of the epigenetic mark provided by the histone H3 variant, centromere protein A (CENP-A), but the molecular mechanisms that underlie its remarkable stability remain unclear. Here, we define the contributions of each of the three candidate CENP-A nucleosome-binding domains (two on CENP-C and one on CENP-N) to CENP-A stability using gene replacement and rapid protein degradation. Surprisingly, the most conserved domain, the CENP-C motif, is dispensable. Instead, the stability is conferred by the unfolded central domain of CENP-C and the folded N-terminal domain of CENP-N that becomes rigidified 1,000-fold upon crossbridging CENP-A and its adjacent nucleosomal DNA. Disrupting the 'arginine anchor' on CENP-C for the nucleosomal acidic patch disrupts the CENP-A nucleosome structural transition and removes CENP-A nucleosomes from centromeres. CENP-A nucleosome retention at centromeres requires a core centromeric nucleosome complex where CENP-C clamps down a stable nucleosome conformation and CENP-N fastens CENP-A to the DNA.

[1] Department of Biochemistry and Biophysics, Perelman School of Medicine, University of Pennsylvania, Philadelphia, Pennsylvania 19104, USA. [2] Graduate Program in Biochemistry and Molecular Biophysics, Perelman School of Medicine, University of Pennsylvania, Philadelphia, Pennsylvania 19104, USA. [3] Whitehead Institute for Biomedical Research and Department of Biology, Massachusetts Institute of Technology, Cambridge, Massachusetts 02142, USA. [4] Ludwig Institute for Cancer Research and Department of Cellular and Molecular Medicine, University of California, San Diego, La Jolla, California 92093, USA. [5] Graduate Program in Bioengineering, University of Pennsylvania, Philadelphia, Pennsylvania 19104, USA. † Present addresses: Norwegian Centre for Molecular Medicine and Department of Chemistry, University of Oslo, Oslo 0349, Norway (N.S.); Institut Curie, PSL Research University, CNRS, UMR 144, 26 rue d'Ulm, Paris F-75005, France (D.F.). Correspondence and requests for materials should be addressed to B.E.B. (email: blackbe@mail.med.upenn.edu).

The centromere is the specialized region of chromatin that directs accurate chromosome segregation in cell division[1,2]. The centromere recruits the proteinaceous kinetochore, which attaches to spindle microtubules during mitosis or meiosis. A model for the epigenetic specification of centromere identity has emerged wherein pre-existing nucleosomes with a histone H3 variant named centromere protein A (CENP-A)[3,4] direct the local assembly of newly synthesized CENP-A[5,6], with CENP-A deposition occurring once per cell cycle following completion of mitosis[7,8]. Critically, this model relies on the stable maintenance of CENP-A nucleosomes at a single site on each chromosome throughout the remainder of the cell cycle.

Indeed, relative to the other H3 variants (that is, H3.1 and H3.3) that turnover in chromatin[9–11], CENP-A experiences essentially no detectable turnover once assembled at a centromere[6,7,9,11], and the stability has been measured out to >1 year where it preserves centromere identity in oocytes that are arrested in a prophase-like state during the entire fertile lifespan of female mice[12]. Particularly in the female germline or any somatic cell types that do not undergo very rapid divisions, maintaining centromere identity between rounds of CENP-A nucleosome assembly is critical for faithful chromosome inheritance. Thus, defining the molecular processes that confer the extraordinary stability of CENP-A nucleosomes is of outstanding interest in chromosome biology.

To date, both intrinsic features (that is, those encoded in the sequence of CENP-A, itself) and extrinsic factors (that is, constitutive centromere components that bind directly to CENP-A nucleosomes) have been considered as candidates that contribute to this distinctive stability. Residues that rigidify the interface between CENP-A and its partner histone, H4, are necessary but not sufficient for this stability[11,13–15], so extrinsic factors must be considered. The only two proteins of the constitutive centromere-associated network (CCAN) known to make specific contacts with CENP-A nucleosomes on all functional mammalian centromeres are CENP-C and the CENP-N subunit of the CENP-L-N complex[16–19]. Between these components of the CCAN, there are a total of three nucleosome-binding domains: two on CENP-C (the central domain [CENP-C$^{CD}$ a.a. 426–537][17] and the CENP-C motif [CENP-C$^{CM}$ a.a. 736-758] (ref. 19)) and one comprised of the N-terminal portion of CENP-N (CENP-N$^{NT}$ a.a. 1–240) (refs 16,18).

For the two nucleosome-binding domains of CENP-C, CENP-C$^{CD}$ and CENP-C$^{CM}$ each are proposed to engage the CENP-A nucleosome through similar histone contact points and without any local secondary structure of their own[19]. CENP-C$^{CD}$ is conserved in mammals[19], was mapped initially as the primary CENP-A nucleosome contact site, and has high specificity for CENP-A nucleosomes versus its counterparts with canonical H3 (ref. 17). CENP-C$^{CD}$ also directs a structural transition of the CENP-A nucleosome that changes the shape of the octameric histone core, slides the gyres of the nucleosomal DNA past one another, and generates both surface and internal rigidity to the histone subunits[11,20]. CENP-C$^{CM}$, on the other hand, is conserved from yeast to humans, and represents the only identified nucleosome-binding domain in species lacking a conserved CENP-C$^{CD}$ (ref. 19). CENP-C$^{CM}$ is the only CENP-A nucleosome-binding domain for which there exists atomic-level structural information, with a crystal structure of it bound to a canonical nucleosome in which the 6 a.a. C-terminal tail of CENP-A replaces the corresponding region of histone H3 (ref. 19). This structure revealed that CENP-C$^{CM}$ uses a so-called 'arginine anchor' to recognize the acidic patch on the H2A-H2B dimer[19]. An arginine anchor is the shared feature of a diverse set of nucleosome-binding proteins studied to date[21–25], establishing an emerging paradigm for nucleosome recognition[26].

Prior reports have suggested that either or both of the nucleosome-binding domains of CENP-C could be important for its own localization to centromeres[17,19,27–30]. CENP-N$^{NT}$ recognizes the CENP-A nucleosome via the CENP-A targeting domain (CATD)[13,16,18,31], but it is not known whether its binding site on the nucleosome extends to other histones in a similar manner as the CENP-C nucleosome-binding domains[19]. While a prior study using labelled CENP-N and CENP-C expressed in reticulocyte extracts suggested that they can coexist on the same nucleosome, there existed a need to use purified components to resolve proposals for CENP-C and CENP-N to bind to the same[17] or different[31,32] CENP-A nucleosomes, and to study the nature of such a combined complex. Depletion of CENP-C reduces CENP-A nucleosome stability[11], but this finding does not delineate between a role for CENP-C$^{CD}$ or CENP-C$^{CM}$. Also, CENP-C depletion leads to partial removal of the CENP-L-N complex[18], so it remains possible that CENP-N$^{NT}$ is responsible for CENP-A nucleosome retention. Thus, it is currently unclear which of the CENP-C or CENP-N domains is important for maintaining centromere identity and the extent to which they may cooperate to stabilize centromeric chromatin.

Here, we define the contributions of each of the three nucleosome-binding domains present within the CCAN for maintaining centromere identity. To do this, we use a combination of gene editing, rapid inducible degradation of centromere components, biochemical reconstitution, hydroxyl radical footprinting and hydrogen/deuterium exchange coupled to mass spectrometry (HXMS). Our data establish an essential core centromeric nucleosome complex (CCNC) that is critical for CENP-A stability and maintenance of centromere integrity.

## Results

**CENP-C$^{CD}$ confers stability to CENP-A nucleosomes.** Upon embarking on our effort to define the molecular processes that confer stability to CENP-A nucleosomes, we first turned our attention to CENP-C. We reasoned that if either or both of the nucleosome-binding domains of CENP-C were indeed required for its localization to centromeres[17,19,27–30], then we could not accurately define which of the domains may confer stability to CENP-A. To define the requirements for these domains in the absence of endogenous CENP-C, we employed a human DLD-1 cell line in which both alleles of CENP-C are tagged with an auxin-inducible degron (AID)[33,34] and EYFP tags[35] (Fig. 1a). In this background, we introduced untagged versions of either wild-type or mutant CENP-C proteins, constitutively expressed from a unique genomic locus (Fig. 1b). The AID-EYFP-tagged CENP-C with an otherwise wild-type protein coding sequence is degraded to below the level of detection within 30 min of addition of the synthetic auxin, indole-3-acetic acid (IAA)(Supplementary Fig. 1a–d). This allowed us to exclusively detect the rescue constructs with an antibody directed against CENP-C (Fig. 1c–e). Since all constructs are expressed at roughly equal levels as the AID-EYFP-tagged version before IAA treatment, there is an expected drop in the amount that is detectable at centromeres after IAA treatment even with the wild-type full-length version [CENP-C(FL); Fig. 1d,e and Supplementary Fig. 1e,f]. The removal of the CD led to partially diminished CENP-C localization, whereas removal of the CM had little effect, even when removed in combination with the CD (Fig. 1d,e). Thus, the CD and CM are not strictly necessary for CENP-C localization, consistent with the fact that CENP-C makes multiple other direct contacts within the meshwork of the CCAN[18,32,36].

Previously, we found that slow reduction of CENP-C (via shRNA treatment) causes a marked decrease in the retention of SNAP-tagged and tetramethylrhodamine-Star (TMR*)

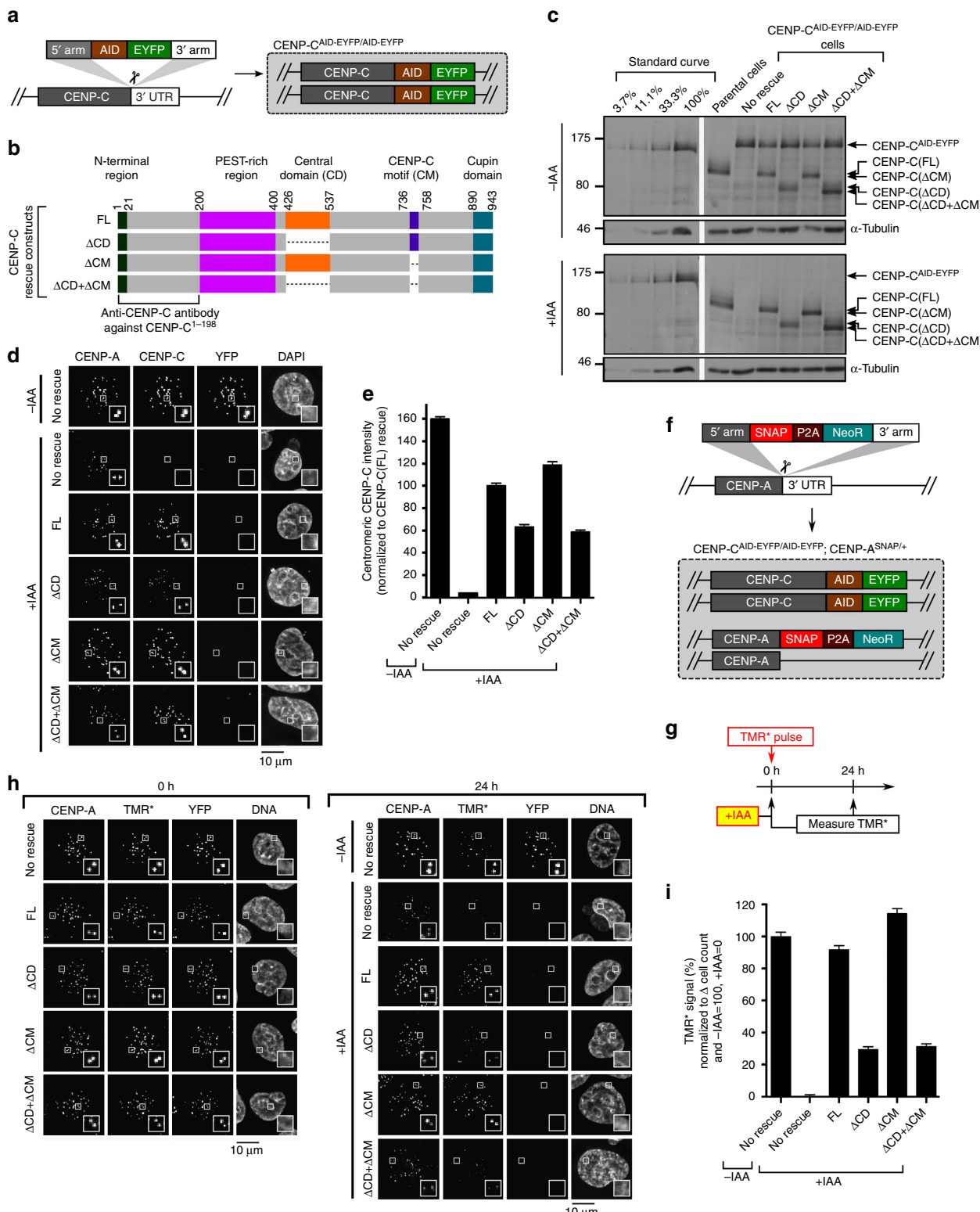

**Figure 1 | CENP-C$^{CD}$ is the only nucleosome-binding domain of CENP-C required for retention of CENP-A nucleosomes.** (**a**) Schematic representation of CENP-C$^{AID-EYFP/AID-EYFP}$ cells. (**b**) Rescue constructs constitutively expressed at unique FRT site. FL, full length. (**c**) Immunoblot of CENP-C$^{AID-EYFP/AID-EYFP}$ cells (with and without 4 h of auxin-induced CENP-C depletion), using an antibody generated against CENP-C (a.a. 1–198). See Supplementary Fig. 9 for uncropped blot. (**d**) Representative images, in which the loss of YFP signal verifies depletion of CENP-C-AID-EYFP after 24 h of IAA, and CENP-C antibody then exclusively detects rescue constructs. Scale bar, 10 μm. (**e**) Quantification of **d**. (**f**) Schematic representation for SNAP-tagging CENP-A at its endogenous locus. (**g**) Schematic representation for pulse-chase experiment, in which CENP-C$^{AID-EYFP/AID-EYFP}$ cells expressing rescue constructs of either CENP-C(FL) or CENP-C domain deletion mutants were pulse-labelled with TMR* and assessed for retention of the existing pool of CENP-A molecules. (**h**) Representative images from experiment diagrammed in **g**. Scale bar, 10 μm. (**i**) Quantification of **h**. See also Supplementary Fig. 1k–m. All graphs are shown as mean ± 95% confidence interval ($n > 2,000$ centromeres in all cases).

pulse-labelled CENP-A at centromeres[11]. This strategy allows us to monitor the pool of CENP-A nucleosomes existing prior to the pulse labeling[7,9,11], so that any effects of CENP-C depletion on nascent CENP-A assembly[17,37,38] do not complicate our analysis. Therefore, we next tested whether these domains of CENP-C are required for the retention of CENP-A at centromeres. We first

added the SNAP tag to endogenous CENP-A using CRISPR-Cas9-mediated genome editing (Fig. 1f and Supplementary Fig. 1g–j), and confirmed that rapid removal of CENP-C-AID-EYFP with no rescue causes a dramatic decrease over 24 h of the existing pool of CENP-A at centromeres (Supplementary Fig. 1k–m). CENP-C(FL) rescued CENP-A retention, whereas

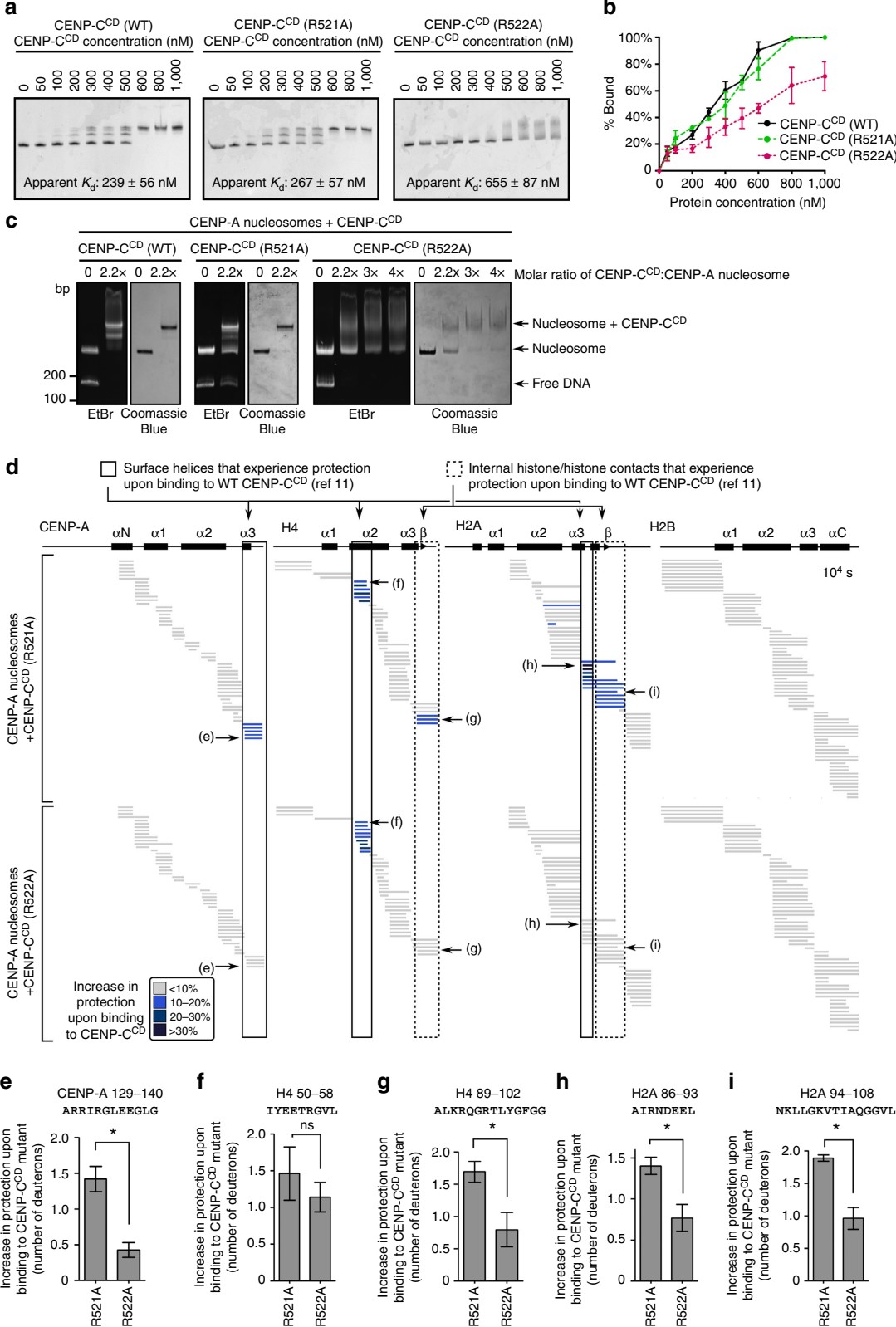

replacement with CENP-C(ΔCD) resulted in a marked decrease in CENP-A retention (Fig. 1g–i). In contrast, CENP-C(ΔCM) did not diminish CENP-A retention (Fig. 1g–i). Therefore, CENP-C[CD] is required for the retention of CENP-A at centromeres.

**The arginine anchor of CENP-C[CD] stabilizes CENP-A.** We next determined whether or not the stability CENP-C[CD] imparts to CENP-A nucleosomes is attributable to the CENP-A nucleosome structural transition[11,20]. We reasoned that by reducing the contact points of the CENP-C[CD] with the H2A-H2B dimer[19] that we could generate a version of CENP-C that could bind to CENP-A nucleosomes but not drive the nucleosome structural transition[11,20] that we predicted would be central to stabilizing CENP-A at centromeres. We chose two adjacent arginines (R521 and R522) within the CENP-C[CD] that are proposed to contact an acidic patch of the nucleosome on the surface of the H2A-H2B dimer[19], and performed quantitative binding studies on CENP-A nucleosomes in which one histone subunit (H2B) is fluorescently labelled at a site (at the position corresponding to K120) distal to the binding surface of CENP-C[CD] (Supplementary Fig. 2a). We reasoned it was likely that existing isothermal calorimetry data[19] (that used canonical nucleosomes with the C-terminal 6 a.a. of CENP-A appended to conventional H3) does not clearly distinguish altered binding from a complete loss of binding; since the complex binding surface for CENP-C[CD] on the nucleosome involves three different histone subunits and a nucleosome structural transition for *bona fide* CENP-A nucleosomes[11,20], complicating the interpretation of thermodynamic measurements. Instead, our native polyacrylamide gel electrophoresis (PAGE) analyses would clearly distinguish unbound, bound and higher-order aggregates that can form with the WT protein at very high concentrations. We find that while the binding affinity of CENP-C[CD] R521A is close to that of WT CENP-C[CD], CENP-C[CD] R522A exhibits a two to three-fold decrease in binding affinity (Fig. 2a,b). The discrete mobility on native PAGE of the CENP-A nucleosome core particle (NCP)-CENP-C[CD] complex is lost in R522A, but preserved in R521A (Fig. 2a,c). Therefore, both mutations preserve the ability to bind to the NCP, but that the R522A may perturb the highly ordered nature of the complex that CENP-C[CD] forms with the NCP. Meanwhile, mutation of one of the hydrophobic residues (W530) proposed to contact the hydrophobic tail of CENP-A abolishes binding to CENP-A nucleosomes (Supplementary Fig. 2b–e).

We then measured the effects of these mutations using HXMS, an approach that revealed HX protection that maps unambiguously to the buried centre of the nucleosome to a region encapsulating a β-sheet that forms between H2A and H4, coinciding with the CENP-A nucleosome structural transition conferred by WT CENP-C[CD] (ref. 11). HX measures the rate of amide proton exchange along the polypeptide backbone of proteins, and protection occurs through stabilization of H-bonds within secondary structure[39] (that is, within histone α-helices or between the β-strands that form with loop L1/L2 contacts between histone pairs like CENP-A and H4 (refs 11,13)) or via direct backbone interactions[39]. Reconstitution of complexes at the concentrations and high nucleosome saturation required for the clear interpretation of HXMS experiments (see Methods) were achieved with WT, R521A and R522A versions of CENP-C[CD] (Fig. 2c). HXMS analysis revealed that CENP-C[CD](R521A) forms a similar complex as wild-type CENP-C[CD], with protection of the surface helices of the CENP-A NCP as well as the interior H2A-H4 interface (Fig. 2d)[11]. On the other hand, although CENP-C[CD](R522A) still binds to CENP-A NCPs (Fig. 2a–c), its mode of binding is grossly perturbed: it still contacts and stabilizes the H4 α2-helix on the surface of the NCP, but the HX protection is reduced at the other surface helices (one each on CENP-A and H2A; Fig. 2d). We interpret these results to mean that the reduced HX protection on the surface of H2A is directly due to the removal of the CENP-C[CD] arginine anchor. The altered dynamics in HX are extended to reduce protection at a contact site with CENP-A (near CENP-C a.a. 530) that lies between the CENP-C[CD] N-terminal (CENP-C a.a. 522) nucleosome contact point (on H2A) and its C-terminal (near CENP-C a.a. 535) contact point (on H4)[19] (Fig. 2d–f,h). Most importantly, removal of the arginine anchor by the R522A mutation leads to loss of HX protection at the H2A-H4 interface at the interior of the nucleosome (Fig. 2d,g,i). Since CENP-C(R522A) binding fails to stabilize the internal H4/H2A interface similar to CENP-C(Δ519–533), which is missing the entire region required for binding to CENP-A nucleosomes, we predicted that both mutants would also fail to confer CENP-A stability. After generating the respective cell lines (Fig. 3a), and adding IAA to remove CENP-C-AID-YFP, all mutants localize to centromeres to a level equivalent to the wild-type protein (Fig. 3b,c). We measured retention of TMR*-labelled CENP-A, and found that both CENP-C(Δ519–533) and CENP-C(R522A) were markedly reduced in their ability to retain CENP-A at centromeres, whereas the CENP-C(R521A) mutation had no effect (Fig. 3d,e). The R522A result is particularly striking, indicating that R522 is the key arginine anchor of CENP-C[CD] and providing a prime example of how an arginine anchor on a nucleosome-binding protein can be a lynchpin for a central biological process such as maintaining centromere identity. Together with our HXMS results, these data strongly indicate that stabilization of the interior of the CENP-A nucleosome requires the H2A-H2B contacts via R522 of CENP-C

**Figure 2 | The arginine anchor of CENP-C[CD] is critical for the CENP-A nucleosome structural transition.** (**a**) Representative native PAGE analysis of CENP-A NCPs harbouring Cy5-labelled histone H2B that have been incubated with the indicated concentrations of CENP-C[CD] (WT or the indicated mutants). Each reaction contains 200 nM nucleosomes. Cy5 fluorescence was detected on a Typhoon phosphorimager, and CENP-C binding retards the mobility. Both WT and R521A show crisp shifts to bands with one or two copies of CENP-C bound to the nucleosome. R522A exhibits a more smeary appearance when bound to the CENP-A nucleosome (see also **c**), and the species with a single molecule of CENP-C[CD](R522A) was not clearly resolved. Listed on the graphs are apparent $K_d$ values for these binding experiments (values shown are mean ± s.d.; $n = 3$). (**b**) Quantification of three independent experiments (values shown are mean ± s.d.) performed as in **a**. Note that for some data points, the error bars are too small to be visible in the graph. (**c**) CENP-A NCPs in complex with WT or mutant CENP-C[CD], as assessed by native PAGE stained with ethidium bromide (EtBr) and then Coomassie Blue. (**d**) HXMS of all histone subunits of the CENP-A NCP from a single timepoint ($10^4$ s), showing regions that exhibit additional protection from HX upon binding of CENP-C[CD](R521A) or CENP-C[CD](R522A). Each horizontal bar represents an individual peptide, placed beneath the schematics of secondary structural elements of the CENP-A nucleosome. When available, we present the data from all measurable charge states of each of the unique peptides (here and in the similarly formatted plots in the experiments presented in Figs 4–6). (**e–i**) Representative peptides from various histone regions, comparing protection from exchange when the nucleosome is bound to CENP-C[CD] R521A versus R522A, showing faithful detection of differences between the two mutants across multiple replicate experiments (plotted as the mean ± s.d.; $n = 3$). Asterisks denotes differences that are statistically significant ($P < 0.05$; Student's $t$-test).

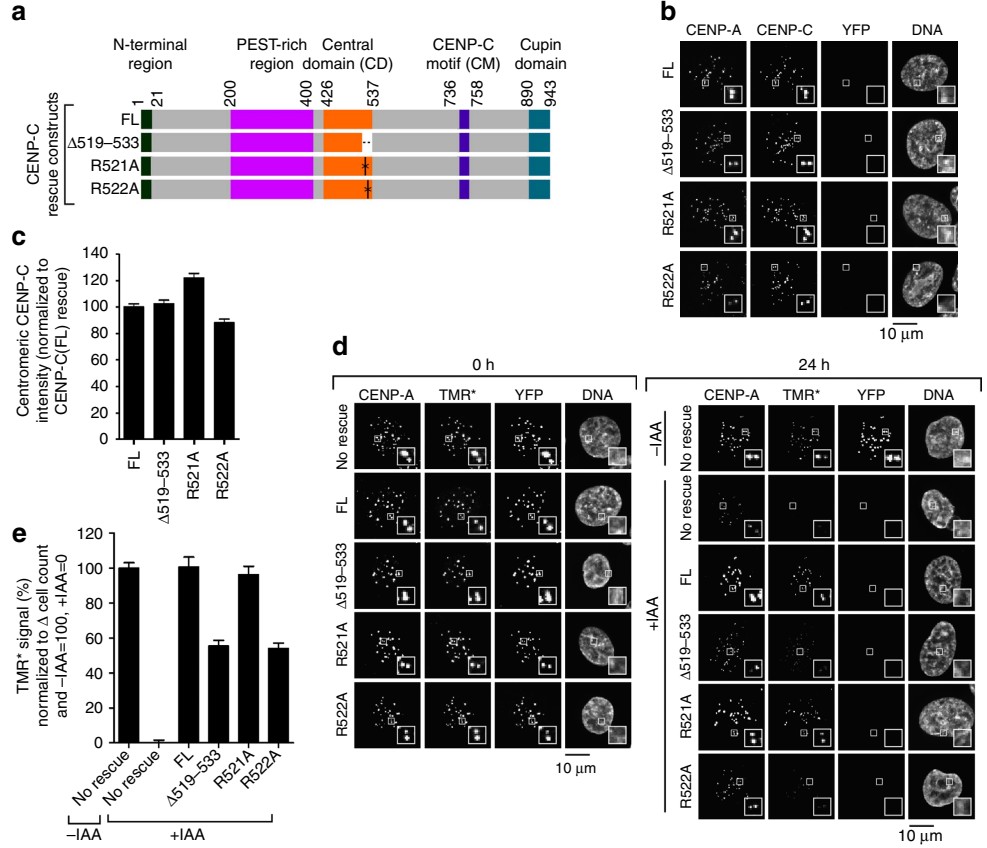

**Figure 3 | The arginine anchor of CENP-C$^{CD}$ is required for CENP-A nucleosome stability at centromeres.** (**a**) Rescue constructs constitutively expressed at the unique FRT site in CENP-C$^{AID-EYFP/AID-EYFP}$ cells. FL, full length. (**b**) Representative images showing localization of CENP-C rescue constructs in CENP-C$^{AID-EYFP/AID-EYFP}$ cells after 24 h of IAA treatment. Scale bar, 10 μm. (**c**) Quantification of **b**. (**d**) Representative images showing CENP-A retention as measured by TMR* assay in cells, similar to schematic representation in Fig. 1g. Scale bar, 10 μm. (**e**) Quantification of **d**. All graphs are shown as mean ± 95% confidence interval ($n > 2000$ centromeres in all cases).

and that the stability of CENP-A nucleosomes due to CENP-C at functional centromeres can be attributed exclusively to the CENP-C$^{CD}$.

**CENP-N$^{NT}$ fastens CENP-A to nucleosomal DNA.** Although a substantial component ($\sim 50\%$) of CENP-A retention at centromeres is attributable to the CENP-A nucleosome structural transition conferred by CENP-C$^{CD}$ (Fig. 3d,e), complete removal of CENP-C leads to a more pronounced defect (Fig. 3d,e and Supplementary Fig. 1k–m), implying that an interacting partner outside of the CENP-C$^{CD}$ also contributes to CENP-A retention. We next considered the CENP-L-N complex because CENP-N$^{NT}$ directly contacts the CENP-A nucleosome[16,18], but additionally requires CENP-C for its centromere localization[18]. Prior work found that the interaction of CENP-L-N with CENP-C occurs in a region (CENP-C a.a. 235–509)[18] overlapping with the CD (CENP-C a.a. 426–537) but outside of the histone contact residues (CENP-C a.a. 519–533). We found that CENP-C$^{235-425}$ binds to CENP-L-N at similar levels to CENP-C$^{235-509}$ (Fig. 4a), and that CENP-C$^{235-352}$ was sufficient for this interaction (Supplementary Fig. 3a). Consistent with this, CENP-C(Δ519–533) almost completely rescues CENP-L-N localization (Fig. 4b,c and Supplementary Fig. 3b,c). Taken together, our findings suggest that CENP-C and CENP-N bind to each other with interaction interfaces that are distinct from their nucleosome interaction interfaces. Thus, in principle, CENP-N$^{NT}$ and CENP-C$^{CD}$ could both

contribute to the stability of CENP-A nucleosomes at human centromeres.

To measure the location and magnitude of the stability conferred by CENP-N$^{NT}$ to the CENP-A NCP, we performed HXMS on the assembled complex (Fig. 4d and Supplementary Fig. 3d,f–j). The only region on the entire NCP where we detected substantial protection from HX in the presence of CENP-N$^{NT}$ is within the CATD[13], at a discrete portion that spans the C-terminal region of the α1-helix and the N-terminal portion of L1 (Fig. 4d–f). This location corresponds to a major surface structural feature unique to CENP-A nucleosomes: a bulge of opposite charge as the same site of canonical nucleosomes containing H3 (refs 15,40,41). The HXMS results are consistent with previous work demonstrating that CENP-N recognizes the CATD[16,18]. The region of HX protection is centred around residues R80 and G81 on CENP-A (Fig. 4d, inset), where mutations disrupt CENP-N binding[31].

We next considered how CENP-N specifically recognizes CENP-A when it is in a histone complex wrapped with DNA and how this might contribute to its function at centromere. We employed a well-established approach for nucleosomes[42–44], recently extended to CENP-A nucleosomes assembled with a synthetic positioning sequence[45] that employs hydroxyl radical-mediated cleavage of DNA. We used the same natural CENP-A nucleosome-positioning sequence from human centromeres[11,46] used in our HXMS experiments, but where it is end-labelled[20] for hydroxyl radical footprinting (Fig. 4g). CENP-A nucleosome positioning is strong enough to readily

detect the expected ∼10 bp periodicity of protection from hydroxyl radical cleavage caused by each superhelical turn of the DNA on the surface of the histone octamer (Fig. 4h). CENP-N$^{NT}$ binding does not alter the phasing, but there is very strong added protection at −21 and −22 nt from the dyad axis of the nucleosome (Fig. 4h,i). This location is immediately adjacent to the bulged L1 of CENP-A that is protected from HX (Fig. 4j). Thus, we envision a continuous

binding surface that spans and crossbridges CENP-A and nucleosomal DNA.

This raised the questions of whether the nucleosome-binding surface of CENP-N$^{NT}$ is an extended, unstructured segment, as in CENP-C$^{CD}$, or a well-folded domain. Fortunately, our HXMS experiments on the CENP-A nucleosome complex with CENP-N$^{NT}$ yielded near complete coverage of both the histone fold domains of each nucleosome subunit (Fig. 4d) and

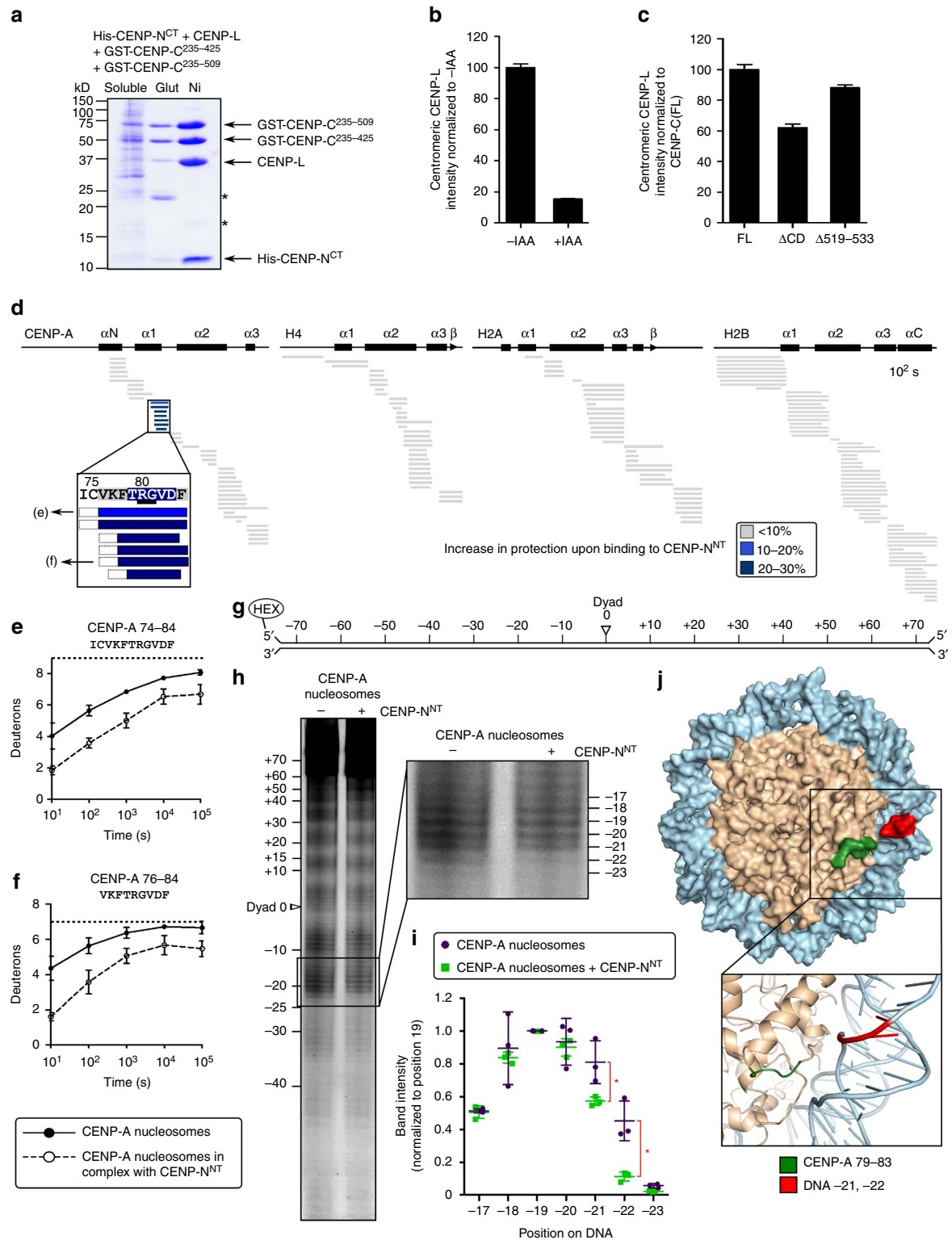

CENP-N$^{NT}$, itself (Fig. 5), for which there is little or nothing known regarding its structure and dynamics. For CENP-N$^{NT}$, we found substantial HX protection for the CENP-N$^{NT}$ molecule alone (Fig. 5), suggesting it is a folded domain. This protection was markedly increased—taking 100–1,000 times as long to reach the same level of HX—upon binding to CENP-A NCPs (Fig. 5). The dramatic protection from HX on CENP-N$^{NT}$ upon binding to the CENP-A NCP extended through its entire N-terminal ~200 a.a (Fig. 5 and Supplementary Fig. 4a–g). Residues ~209–240 were disordered both before and after binding to CENP-A NCPs (Fig. 5a and Supplementary Fig. 4h–i), indicating that this region is unlikely to be involved in binding. Indeed, a further truncation of CENP-N$^{1-205}$ retained the ability to bind to CENP-A nucleosomes (Supplementary Fig. 4j). Since there are key residues for this interaction (R11 and R196)[16] at each end of this domain, it is likely that the folded nature of the CENP-N$^{NT}$ brings together key residues that form the binding surface with CENP-A NCPs. Indeed, important residues for CENP-A nucleosome binding are found at various locations across this region of CENP-N[16].

**The core centromeric nucleosome complex (CCNC).** Since the HX protection on CENP-A NCPs from CENP-N$^{NT}$ is discrete (Fig. 4d) at a nucleosomal surface contact point that remains accessible after CENP-C$^{CD}$ binding[11,19], it seemed reasonable to reconstitute nucleosome complexes with both domains bound simultaneously. Using established conditions that generate a complex with one copy of CENP-C$^{CD}$ bound to each face of the nucleosome[11,19], we added increasing amounts of CENP-N$^{NT}$ (Fig. 6a). We observed a concentration-dependent and stepwise formation of complexes where one and two copies of CENP-N$^{NT}$ bound to the CENP-A NCP complex containing two copies of CENP-C$^{CD}$ (Fig. 6a,b). The complexes were stable through native PAGE analysis, and the dominant species contained equimolar amounts (that is, two copies each) of each core histone (CENP-A, H4, H2A and H2B), CENP-C$^{CD}$, and CENP-N$^{NT}$ (Fig. 6b). We term this complex the CCNC (Fig. 6c and Supplementary Fig. 5a,c). The complex was purified by preparative native PAGE (Supplementary Fig. 5c) and is also stable through sucrose gradient (Supplementary Fig. 5d,e), indicating that the CCNC is stable throughout the lengthy (several hours) separation, even with no gel matrices involved whatsoever.

CENP-A nucleosomes within the CCNC experience protection from HX at multiple sites (Fig. 6e,f and Supplementary Fig. 5b,f–i) corresponding to the additive contributions of CENP-C$^{CD}$ (ref. 11) and CENP-N$^{NT}$ (Fig. 4d–f). Furthermore, the rigidity

conferred to CENP-N is measured out to 100,000 s of exchange, and exhibits clear EX2 behaviour at all timepoints—without any evidence of bimodal peaks or any other fast exchanging species that could have corresponded to a substantially populated unbound, unprotected state—thus providing unambiguous evidence that the complex is stable in solution even on timescales of ~28 h (Fig. 6d). The CCNC exhibits surface protection on CENP-A, H4 and H2A and protection at the internal interhistone H2A-H4 β-sheet that are all conferred by CENP-C$^{CD}$ (ref. 11), as well as the surface bulge protection at the α1-helix and L1 conferred by CENP-N$^{NT}$ (Fig. 6e,f and Supplementary Fig. 5g–i). The discrete HX protection pattern emphasizes the specific nature of CCNC assembly in solution. CENP-C$^{CD}$, itself, undergoes rapid HX, consistent with CENP-C$^{CD}$ existing as a primarily linear polypeptide lacking defined secondary structure[19], although there is reduced HX at the earliest timepoints within ~a.a. 515–537 when bound to CENP-A nucleosomes (Supplementary Fig. 6). Within the CCNC, CENP-N$^{NT}$ still experiences massive slowing of HX across most of its folded nucleosome-binding domain, with exception of its a.a. 99–122 region, suggesting that its mode of nucleosome binding could be altered at that specific location by the co-presence of CENP-C$^{CD}$ (Fig. 6d and Supplementary Fig. 7). Thus, in the context of the CCNC, each non-histone subunit acts in a complementary way to stabilize the particle: CENP-C$^{CD}$ binding directs a structural transition of the nucleosome that stabilizes the interior of the octameric histone core and stabilizes surfaces helices on three of the histone subunits (Fig. 6e,f), whereas CENP-N$^{NT}$ stabilizes the CENP-A surface bulge on the nucleosome surface (Fig. 6e,f) and crossbridges it to the adjacent DNA (Fig. 4g–i).

**CENP-A nucleosome stability requires CENP-C and CENP-N.** Our finding that a stable CCNC can be assembled from its component parts (Fig. 6) supports the notion that CENP-N provides stability to centromeric chromatin that cannot be attributed to CENP-C$^{CD}$ (Figs 1 and 3). To test this notion, we focused our analysis back on our cell lines where the levels of the two components can be modulated. CENP-C and CENP-N display partially interdependent localization to centromeres in human cells[18], complicating the analysis of their interactions with CENP-A. During the 24 h timescale, in which we measure CENP-A retention at centromeres, CENP-C removal also leads to loss of most but not all CENP-L-N (Fig. 4b and Supplementary Fig. 3b). CENP-N removal, using a similar AID-tagging approach of both CENP-N alleles[18], reduces CENP-C levels at centromeres by ~half (Supplementary Fig. 8a–c). Interestingly, CENP-N-AID

**Figure 4 | CENP-N$^{NT}$ crossbridges CENP-A to DNA.** (**a**) Coomassie Blue-stained SDS–PAGE of co-purification with described protocol[18] of CENP-L/His-CENP-N$^{CT}$ with GST-CENP-C$^{235-509}$ and GST-CENP-C$^{235-425}$ by glutathione-agarose (Glut) or Nickel-NTA-agarose (Ni). (**b**) Localization of CENP-L-N in CENP-C$^{AID-EYFP/AID-EYFP}$ cells before and after 24 h of IAA treatment, assessed using anti-CENP-L[18] (Supplementary Fig. 3b for images). (**c**) Localization of CENP-L-N in CENP-C$^{AID-EYFP/AID-EYFP}$ cells constitutively expressing the rescue constructs CENP-C(FL), CENP-C(ΔCD) or CENP-C(Δ519–533), after 24 h of IAA treatment. (Supplementary Fig. 3c for images) All graphs are shown as mean ± 95% confidence interval ($n > 2,000$ centromeres in all cases). (**d**) HXMS of all histone subunits of the CENP-A NCP from a single timepoint ($10^2$ s), showing protection at CENP-A(79–83) upon binding to CENP-N$^{NT}$. The first two residues of each peptide are boxed in dashed black lines because exchange of the first two backbone amide protons cannot be measured[64]. (**e,f**) Representative peptides spanning the CENP-A surface bulge over the timecourse. The maximum number of deuterons possible to measure by HXMS for each peptide is shown by the dotted line. All peptides are plotted at every timepoint as mean ± s.d. from triplicate experiments. Note that for some data points, the error bars are too small to be visible in the graph. (**g**) Schematic representation of the 5′-fluorescently labelled 147 bp α-satellite DNA sequence used in footprinting experiments. (**h**) Representative hydroxyl radical footprinting experiment of CENP-A nucleosomes vs. CENP-A nucleosomes in complex with CENP-N$^{NT}$, with inset showing magnification of positions −17 to −23. (**i**) Quantification of band intensities from three independent experiments, shown as mean ± s.d. normalized to DNA position −19 (this position was chosen because it was expected to be very exposed for hydroxyl radical-mediated cleavage with and without CENP-N$^{NT}$). Asterisks denotes differences that are statistically significant ($P < 0.05$; Student's $t$-test). (**j**) A molecular model of the CENP-A nucleosome (PDB 3AN2)[41], in which the DNA sequence was modified[20] to that used in the footprinting experiment: CENP-A a.a. 79 − 83 is labelled in green, and DNA positions − 21 and − 22 are labelled in red.

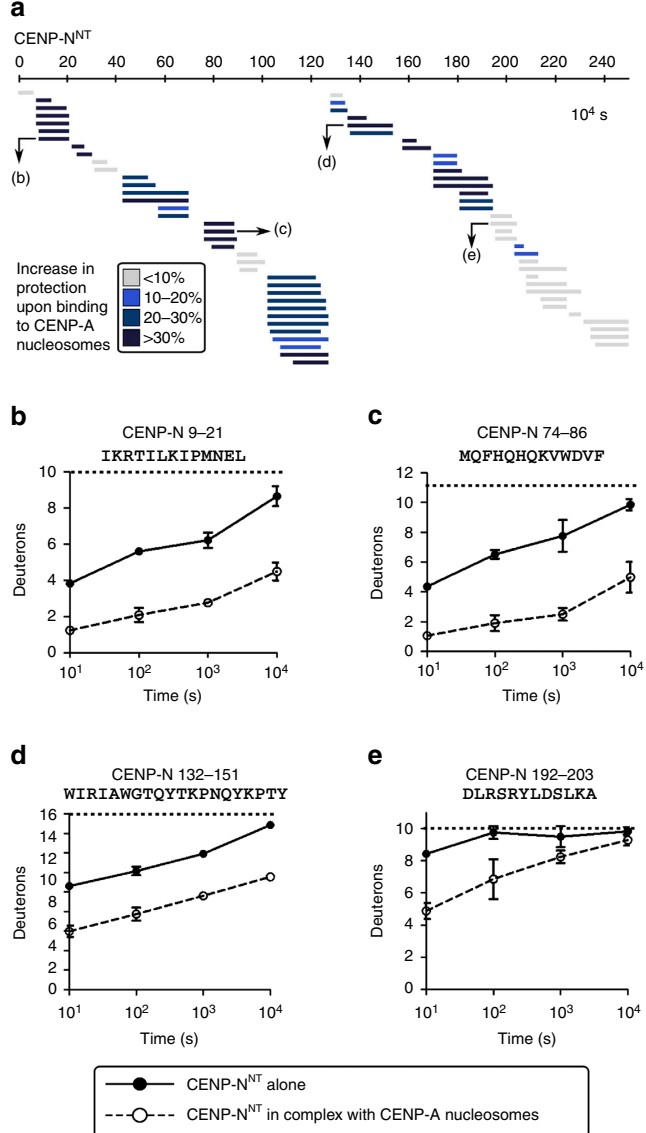

**Figure 5 | CENP-N^{NT} undergoes global stabilization upon binding to the CENP-A nucleosome.** (**a**) HXMS of CENP-N^{NT} from a single timepoint ($10^4$ s), showing substantial protection from HX spanning the $\sim 200$ a.a. domain upon binding to CENP-A NCP. (**b–e**) Representative peptides spanning CENP-N^{NT} over the timecourse. All peptides are plotted at every timepoint as mean ± s.d. from triplicate experiments. Note that for some data points, the error bars are too small to be visible in the graph.

removal produces a pronounced defect in CENP-A retention (Fig. 7a–c), suggesting a direct role of CENP-N in CENP-A retention, since reducing CENP-C levels by half is unlikely to be responsible for the full magnitude of this effect. Importantly, combined removal of CENP-N (by siRNA treatment) and CENP-C (by IAA treatment) had an additive effect, severely compromising CENP-A retention (Fig. 7d–f and Supplementary Fig. 8d,e). This indicates that the low level of CENP-L-N remaining after 24 h of CENP-C depletion (Fig. 4b and Supplementary Fig. 3b) accounts for the residual stability to CENP-A nucleosomes not accounted for by the CENP-C^{CD} alone (Fig. 3d,e). More importantly, this supports a model wherein CENP-C and CENP-N are roughly equal partners necessary to form the CCNC and maintain CENP-A nucleosome levels at centromeres (Fig. 8).

## Discussion

Our physical studies of CENP-A nucleosome complexes combined with gene replacement and rapid depletion of the non-histone CCAN proteins, CENP-C and CENP-N, provide the molecular basis for the extraordinary stability of CENP-A nucleosomes that is at the heart of the epigenetic mechanism that maintains the identity of centromere location on every chromosome. At steady-state, we envision that the relevant nucleosome required to maintain centromere identity has $\sim 147$ bp of DNA wrapped around an octameric histone core containing two copies each of CENP-A, H4, H2A, H2B and two copies each of CENP-C and CENP-N. CENP-C-binding confers the most pronounced physical changes in the CENP-A nucleosome structural transition that alters nucleosome shape, enhances the tendency of CENP-A nucleosomes to sample states with 20 bp of DNA unwrapping at each nucleosome terminus, and confers internal and surface rigidity to the histone core[11,20]. CENP-N, while having a more discrete impact on the histone core of the NCP (Fig. 4), has a binding site that crossbridges a key DNA contact point on the NCP to the (CENP-A/H4)$_2$ heterotetramer (Fig. 4g–j). We expect that normally these are the direct chromatin contacts at the interface with the kinetochore, including those recently reconstituted with purified components[47]. Although, since contacts between CENP-A and the other CCNC components can be bypassed in mitosis[48], CCNC function may be more relevant to maintaining centromere identity during the remainder of the cell cycle.

Independent recognition of a single nucleosome by two different chromatin components, as we find occurs within the CCNC, has not been well studied in any chromatin context. The small but growing list of physical studies of nucleosome-recognition proteins[21–25] uniformly involves a key contact point between an arginine anchor with the nucleosomal acidic patch[26]. Three previous studies had claimed that mutation of R522 (or its corresponding position in *Xenopus* CENP-C) disrupts centromere targeting of CENP-C, but none provided a definitive answer for mammalian CENP-C: one study was done in *Xenopus* extracts[17], and the two studies in human cells used truncated CENP-C transgenes that were overexpressed[19,49]. Our study advances the field in part because it interrogates the nucleosome-binding domains of CENP-C in a gene replacement system, using one in which the endogenous CENP-C is rapidly and completely removed, and the replacement CENP-C constructs are untagged, full-length, and expressed at near endogenous levels (Fig. 1c). More broadly, our findings show a remarkable role for an arginine anchor beyond their established role in nucleosome recognition[26] to a role in altering nucleosome shape and function. R522A preserves the ability of CENP-C to bind to CENP-A nucleosomes (Fig. 2a–c) and accumulate at centromeres (Fig. 3b,c), but we pinpoint a role for R522 for CENP-A maintenance at the centromere (Fig. 3d,e), driving the nucleosome structural transition that stabilizes the interior of the CENP-A nucleosome (Fig. 2d,e). It is possible that mutation of another residue within CENP-C^{CD} could also retain binding, while compromising the CENP-A nucleosome structural transition, but disruption of the R522 arginine anchor in our gene replacement systems indicates that this common feature in diverse nucleosome-binding proteins can play an important functional role, beyond the role of molecular recognition.

CENP-C^{CD} is particularly remarkable because its high specificity for CENP-A nucleosomes is mediated by a very small feature (the 6 a.a. C-terminal tail)[16], but its binding confers stabilization that spreads throughout much of the octameric core of the nucleosome as well as to the position of the DNA gyres[11,20]. Remarkably, CENP-C^{CD} does this without having any defined secondary structure of its own.

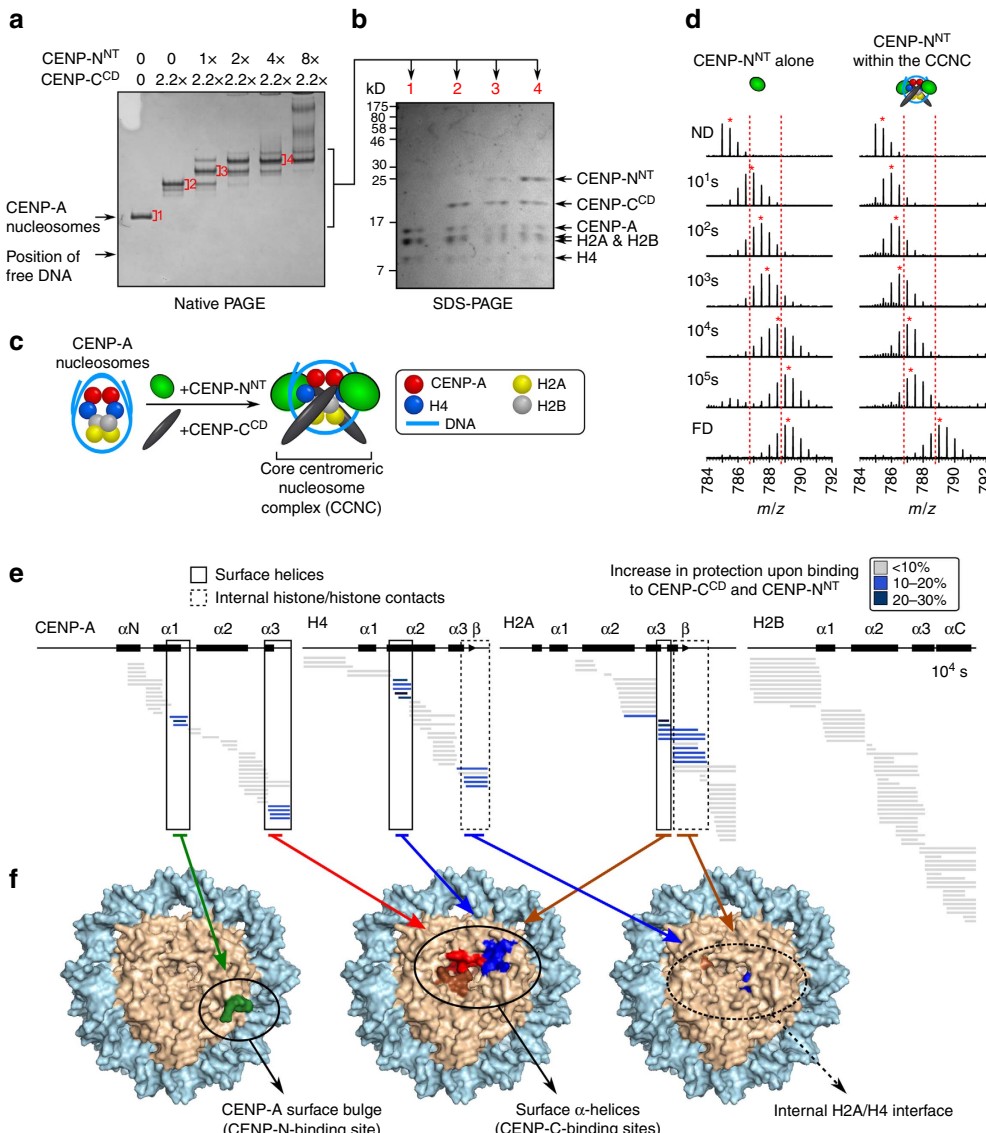

**Figure 6 | CENP-C$^{CD}$ and CENP-N$^{NT}$ simultaneously bind to the same CENP-A NCP and generate internal and surface stability.** (**a**) Coomassie Blue-stained native PAGE of binding reactions with CENP-N$^{NT}$ and CENP-C$^{CD}$ and CENP-A NCPs. (**b**) Indicated bands from native PAGE excised and run on SDS–PAGE. (**c**) Schematic representation of formation of the CCNC. (**d**) A representative peptide of CENP-N$^{NT}$ (a.a. 9 − 21) that shows substantial protection upon binding to CENP-A nucleosomes. The peptide is shown from CENP-N$^{NT}$ alone (left) versus as part of the CCNC (right). Dotted red lines serve as guideposts to highlight the differences in *m/z* shifts between the two samples. A red asterisk denotes the centroid location of each peptide envelope, and the numerical value in blue indicates the centroid mass of the peptide envelope. It is important to note that this peptide exhibits clear EX2 behaviour at all timepoints when part of the CCNC (without any evidence of bimodal peaks), indicating that this complex is stable in solution even on timescales of 100,000 s (∼28 h). (**e**) HXMS of all histone subunits of the CENP-A NCP from a single timepoint ($10^4$ s). (**f**) Regions showing substantial protection from HX mapped onto the structure of the CENP-A NCP (PDB ID 3AN2)[41]. (left) The exposed CENP-A bulge, to which CENP-N binds. (middle) The surface helices to which CENP-C binds. (right) Internal histone–histone contacts that undergo stability upon CENP-C binding.

Most aspects of the mechanism used by CENP-N$^{NT}$ contrast starkly with that used by CENP-C$^{CD}$. The only notable similarity is that CENP-N$^{NT}$ uses a small feature on the surface of the CENP-A nucleosome to achieve its high specificity of binding to CENP-A nucleosomes. In contrast to the widespread HX protection conferred to the NCP by CENP-C$^{CD}$, the only HX protection we observed with CENP-N$^{NT}$ maps to loop L1 (Fig. 4d–f). Our findings provide clear biophysical evidence that CENP-N$^{NT}$ recognizes NCPs without accessing the acidic patch on H2A-H2B at all, making it unique relative to other nucleosome-recognition domains studied to date[21–25] and leaving open that site on the NCP for CENP-C$^{CD}$ to bind. CENP-N$^{NT}$ itself is a folded domain (again, in contrast to CENP-C$^{CD}$) even

prior to engaging the CENP-A NCP (Fig. 5 and Supplementary Figs 4 and 7). Its discrete contact points on Loop 1 of CENP-A and the adjacent nucleosomal DNA 21-22 bp from the dyad axis of symmetry stabilizes its own secondary structure (Fig. 5 and Supplementary Figs 4 and 7). Therefore, the combination of previous work[11,16,17,19,20,31] and the work presented here shows that CENP-C$^{CD}$ and CENP-N$^{NT}$ defy expectations in that the 'unfolded' one (CENP-C$^{CD}$) generates substantial structural changes in the NCP (Fig. 6e,f and refs 11,19), whereas the 'folded' one (CENP-N$^{NT}$) changes core histone dynamics only very locally at the points of contact with CENP-A and its adjacent nucleosomal DNA (Figs 4d–j and 6e,f). Furthermore, our combined HXMS and hydroxyl radical footprinting shows that

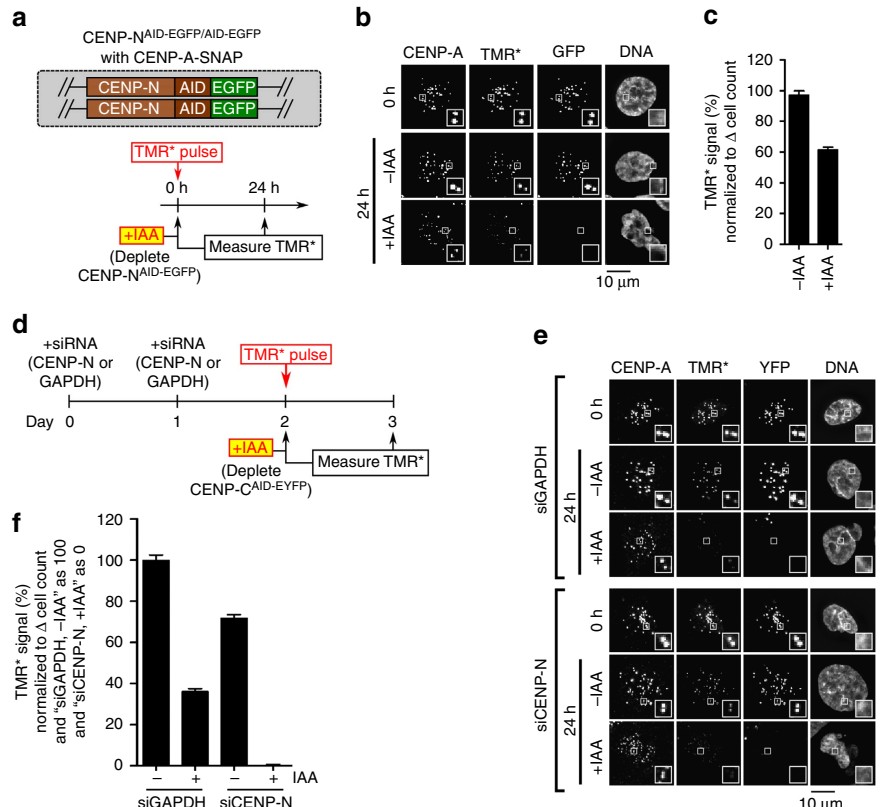

**Figure 7 | CENP-C and CENP-N collaborate to maintain CENP-A nucleosomes at centromeres.** (**a**) Schematic for experiment in which CENP-N$^{AID-EGFP/AID-EGFP}$ cells expressing CENP-A-SNAP at a unique FRT site were pulse-labelled with TMR* and assessed for retention of the existing pool of CENP-A molecules. (**b**) Representative images from experiment diagrammed in **a**. Scale bar, 10 μm. (**c**) Quantification of **b**. (**d**) Schematic representation of experiment, in which CENP-C$^{AID-EYFP/AID-EYFP}$ cells were treated with siCENP-N or siGAPDH and pulse-labelled with TMR*, and the relative CENP-A-SNAP signals were analysed after 24 h (with or without CENP-C depletion by IAA treatment). (**e**) Representative images from experiment described in **d**. Scale bar, 10 μm. (**f**) Quantification of **e**. The value of the 'siGAPDH, − IAA' condition is normalized as 100%, and the value of the 'siCENP-N, + IAA' condition is normalized as 0%. All graphs are shown as mean ± 95% confidence interval ($n > 2,000$ centromeres in all cases).

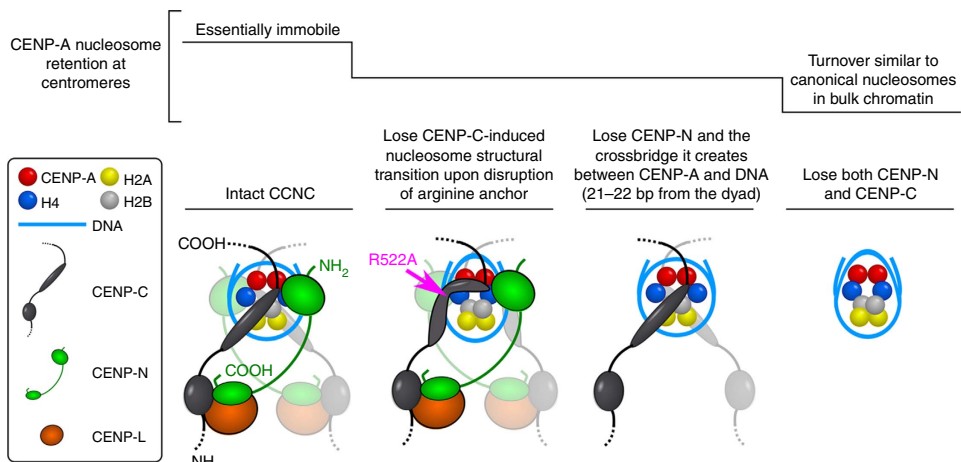

**Figure 8 | Model of the physical basis for the stability of CENP-A nucleosomes within the CCNC.** See text for the details of our model of centromere maintenance. We note that CENP-C$^{CD}$ is shown as an elongated oval that represents a structured loop that has no conventional secondary structure, despite having been historically called a 'domain'. Also, a flexible linker is shown between CENP-C$^{CD}$ and the CENP-C contact point with the CENP-L-N complex, in line with proposals that CENP-C largely exists as an extended and unfolded protein that may span >100 nm at mitotic kinetochores[65–67]. In addition, it is also not known if there is a fixed or variable distance at centromeres from the CENP-C-L-N contact point to the CENP-A nucleosome. It is also unclear if this contact point on CENP-C with CENP-L-N is a folded domain or if it contacts one or both subunits of CENP-L-N.

CENP-N fastens CENP-A to its adjacent DNA, providing an example of a crossbridging mechanism for maintaining nucleosome-encoded epigenetic information that perhaps represent a more general mode for maintaining nucleosome-encoded epigenetic information involving other histone variants (or post-translationally modified canonical histones).

A steady-state CCNC complex required to faithfully maintain CENP-A retention at centromeres does not necessitate that all components exhibit matched turnover rates, themselves. H2A-H2B dimers can come on and off through partial disassembly of nucleosomes, as with canonical nucleosomes. CENP-C and CENP-N could similarly exchange, and indeed both proteins display dynamic behaviours at centromeres[12,50,51], with CENP-N varying in quantity at the centromere depending on the cell cycle stage[18,31,51]. We note that H2A-H2B are not nearly as stable at centromeres as CENP-A and H4 (refs 9,11). The binding mode for CENP-N$^{NT}$ suggests an explanation for this: CENP-C$^{CD}$ binding protects all of the core histones from dissociating from DNA, but CENP-N$^{NT}$ would only protect CENP-A and H4. The hydrophobic stitches between CENP-A and H4 themselves provide yet another required feature to rigidify the particle and maintain it at centromeres[13–15].

In conclusion, our findings demonstrate that the individual CENP-C$^{CD}$ and CENP-N$^{NT}$ subunits independently bind to the nucleosome with non-overlapping effects on the stability and/or shape of the nucleosome (Fig. 8). When both proteins are present, they impart additive effects on the physical properties of CENP-A nucleosomes. At any given time, there are multiple molecules of CENP-C and CENP-N present at the centromere—both directly bound to the CENP-A nucleosomes and directly bound to each other—in a manner that locks in centromere location. By tying faithful inheritance of chromosomes to an epigenetic mark in which the CCNC acts as the fundamental repeating unit, mammals have evolved a remarkably resilient form of chromatin.

## Methods

**Generation of cell lines.** Using DLD-1 Flp-In T-Rex cells stably expressing Tir1 (ref. 34) with CENP-C$^{AID-EYFP/AID-EYFP}$ (ref. 35) as a starting point, endogenous CENP-A was tagged with C-terminal SNAP using CRISPR-Cas9-mediated genome engineering. The sgRNA was designed to target the 3′UTR of CENP-A. The oligonucleotides (5′-CACCGCTGACAGAAACACTGGGTGC-3′ and 5′-AAACGC ACCCAGTGTTTCTGTCAGC-3′) were annealed and inserted into pX330 (Addgene #42230) which already contains Cas9 (ref. 52). To generate the repair template, the SNAP-3xHA sequence[7] followed by a viral 2A peptide[53] and the neomycin resistance gene was synthesized as a gBlock (Integrated DNA Technologies), and 5′ and 3′ homology arms of ~800 bp each were amplified from DLD-1 genomic DNA by PCR. All three pieces were inserted into a pUC19 backbone using HiFi DNA Assembly (NEB). The repair template and pX330 were co-transfected with Lipofectamine 2000 (Invitrogen) in 9:1 ratio, and selected after 5 days using 750 µg ml$^{-1}$ G418. To isolate monoclonal cell lines, cells were subject to limiting dilution after G418 selection. To screen clones, PCR of genomic DNA was performed for every clone (using primers 5′-CCTTCCCCACTCCTTCACAGGC-3′ and 5′-CCTGTGAAAGAGGATGAGCTTACC-3′); insertion of the SNAP tag results in a PCR product of 2243 bp (whereas the PCR product is 614 bp if the allele is unmodified) (Supplementary Fig. 1g–j). Clones containing SNAP-tagged CENP-A were further validated by immunoblotting and TMR* visualization. Stable cell lines constitutively expressing CENP-C rescue constructs were generated by Flp/FRT recombination. Domain deletions and point mutants of CENP-C were generated by PCR site-directed mutagenesis, and the sequences of all constructs (WT and mutant versions) were validated by DNA sequencing. CENP-C constructs were inserted into a pcDNA5/FRT vector and co-transfected with pOG44 (Invitrogen), a plasmid expressing the Flp recombinase, into cells with Lipofectamine 2000 (Invitrogen) according to manufacturer's instructions. Following selection in 400 µg ml$^{-1}$ Hygromycin B, colonies were pooled into polyclonal cell lines. CENP-N$^{AID-EGFP/AID-EGFP}$ cells expressing CENP-A-SNAP were also generated by Flp/FRT recombination: CENP-A-SNAP was inserted into a pcDNA5/FRT vector and co-transfected with pOG44 into CENP-N$^{AID-EGFP/AID-EGFP}$ cells[18] and selected with Hygromycin B as described above.

**Cell culture.** The indicated DLD-1 derivatives described above were cultured in DMEM supplemented with 10% fetal bovine serum, 100 U ml$^{-1}$ penicillin and 100 µg ml$^{-1}$ streptomycin. All cell lines were maintained with 2 µg ml$^{-1}$ puromycin (Sigma). Cell lines in which CENP-A is SNAP-tagged by CRISPR/Cas9-mediated genome editing were maintained with 750 µg ml$^{-1}$ G418. Cell lines containing CENP-C rescue constructs introduced by Flp/FRT recombination were maintained with 400 µg ml$^{-1}$ Hygromycin B. CENP-N$^{AID-EGFP/AID-EGFP}$ cells with CENP-A-SNAP at the FRT site were maintained in 300 µg ml$^{-1}$ G418 and 400 µg ml$^{-1}$ Hygromycin B. To induce degradation of AID-tagged CENP-C or CENP-N, IAA (Sigma) was prepared in water and added to cells at 500 µM for the indicated amounts of time.

**Immunoblotting.** Samples derived from whole cell lysates were separated by SDS–PAGE and transferred to a nitrocellulose membrane for immunoblotting. Blots were probed using the following primary antibodies: rabbit anti-CENP-C (1.7 µg ml$^{-1}$) (ref. 54), mouse mAb anti-α-tubulin (1:4,000, Sigma-Aldrich #T9026), or human anti-centromere antibodies (2 µg ml$^{-1}$, Antibodies Incorporated #15-235). The blots were subsequently probed using the following horseradish peroxidase-conjugated secondary antibodies: Donkey Anti-Human IgG (1:10,000, Jackson ImmunoResearch Laboratories #709-035-149), Amersham ECL Mouse IgG (1:2,000, GE Life Sciences #NA931), Amersham ECL Rabbit IgG (1:2,000, GE Life Sciences #NA934V). Antibodies were detected by enhanced chemiluminescence (Thermo Scientific). Please refer to Supplementary Fig. 9 for the uncropped blots of Fig. 1c and Supplementary Fig. 1d. Note that Supplementary Fig. 1j is not cropped.

**SNAP labelling experiments.** DLD-1 cells were pulse-labelled with 2 µM TMR* (NEB) in complete medium for 15 min at 37 °C, washed with PBS and incubated in the culture medium for 2 h to allow excess TMR* to diffuse out of cells. Cells were then either fixed immediately (for the '0 h' timepoint), or cultured for another 24 h in the presence or absence of 500 µM IAA to induce degradation of the AID-tagged CCAN protein (for the '24 h' timepoints). Cell number was also determined at these timepoints using a haemocytometer, so that the total level of CENP-A turnover could be calculated, as described[9,11]: CENP-A turnover was calculated as [(TMR* intensity at 24 h)/(Avg TMR* intensity at 0 h)]*(Change in cell number). siRNA knockdown of CENP-N was performed as described[55]. Briefly, cells were treated with 20 µM CENP-N siRNAs (siGENOME SMARTpool; Dharmacon, GE Life Sciences #M-015872-02-0005) or GAPDH siRNAs (ON-TARGETplus GAPD Control; Dharmacon, GE Life Sciences #D-001830-01-05).

**Immunofluorescence and microscopy.** For experiments involving CENP-A, CENP-C or CENP-T immunofluorescence, DLD-1 cells were fixed in 4% formaldehyde for 10 min at room temperature and quenched with 100 mM Tris (pH 7.5) for 5 min, followed by permeabilization using PBS containing 0.1% Triton X-100. For experiments involving CENP-L immunofluorescence, DLD-1 cells were pre-extracted with PBS containing 0.1% Triton X-100 for 30 s, fixed with 4% formaldehyde for 10 min and quenched with 100 mM Tris (pH 7.5) for 5 min. All coverslips were then blocked in PBS supplemented with 2% fetal bovine serum, 2% bovine serum albumin and 0.1% Tween before antibody incubations. The following primary antibodies were used: mouse mAb anti-CENP-A (1:1,000, Enzo Life Sciences #ADI-KAM-CC006-E), rabbit pAb anti-CENP-C (1.7 µg ml$^{-1}$; ref. 54), rabbit pAb anti-CENP-T (1 µg ml$^{-1}$; ref. 56) and rabbit pAb anti-CENP-L (1:1,000)[18]. Secondary antibodies conjugated to fluorophores were used: Cy3 Goat anti-Rabbit (1:200, Jackson ImmunoResearch Laboratories #111-165-144) and Cy5 Donkey anti-Mouse (1:200, Jackson ImmunoResearch Laboratories #715-175-151). Samples were stained with DAPI before mounting with VectaShield medium (Vector Laboratories). Images were captured at room temperature on an inverted fluorescence microscope (DMI6000 B; Leica) equipped with a charge-coupled device camera (ORCA AG; Hamamatsu Photonics) and a × 40 oil immersion objective. Images were collected as 0.59 µm z-sections and subsequently deconvolved using identical parameters. To quantify fluorescence intensity of centromeres, the CraQ macro[57] was run in ImageJ with standard settings using DAPI and total CENP-A staining as the reference channel to define regions of interests for quantification of TMR* intensity. To display representative cell images, the z-stacks were projected as single two-dimensional images and assembled using ImageJ (NIH). One representative experiment is displayed from two or more independent experiments. At least 2000 centromeres were analysed for each timepoint.

**Recombinant protein purification.** Human histones and CENP-A were prepared as described[15,58]. Briefly, histones H2A and H2B are expressed as monomers in inclusion bodies and purified under denaturing conditions, then refolded into H2A-H2B dimers. (CENP-A/H4)$_2$ is expressed off of a bicistronic construct as a soluble heterotetramer and purified by hydroxyapatite column followed by cation exchange. Recombinant human CENP-C$^{CD}$ consisting of the central domain (a.a. 426–537) was expressed from a plasmid kindly provided by A. Straight (Stanford)[11,17]. CENP-C is expressed as a GST fusion protein and affinity-purified on a glutathione column. GST is then cleaved by PreScission protease and separated from CENP-C by cation exchange[11,17,58]. PCR site-directed mutagenesis was performed to generate CENP-C$^{CD}$(R521A) and CENP-C$^{CD}$(R522A), and they were expressed and purified using the same protocol as wild-type CENP-C$^{CD}$.

Recombinant human CENP-N[NT]-His was purified with a protocol adapted from a previous study[18]: CENP-N[NT]-His was grown in BL21(DE3)pLysS cells for 6 h at 18 °C, and purified on a 1 ml HisTrap FF column (GE Healthcare), with elution buffer of 50 mM sodium phosphate pH 8.0, 500 mM NaCl, 250 mM imidazole, 1 mM βME and 50% glycerol. PCR-directed mutagenesis was performed to generate the further truncated construct, CENP-N[1–205]-His, and it was purified with the same protocol as CENP-N[NT]-His. Sequential purifications of complexes co-expressing GST- and His-tagged subunits were performed as described[18]. Briefly, complexes were first purified on Ni-agarose, and the elution was bound to glutathione agarose, washed three times, and eluted.

**Assembly of NCPs and complexes.** Six identical repeats of a 147 bp DNA sequence derived from a α-satellite sequence from the human X chromosome[59] was cloned into a pUC57 backbone, with each repeat separated by an EcoRV site. The sequence of each repeat is 5′-ATCAAATATCCACCTGCAGATTCTACCA AAAGTGTATTTGGAAACTGCTCCATCAAAAGGCATGTTCAGCTCTGTGA GTGAAACTCCATCATCACAAAGAATATTCTGAGAATGCTTCCGTTTGCC TTTTATATGAACTTCCTCGAT-3′. This sequence corresponds to the major binding site that the CENP-A nucleosome occupies on human centromeres[46]. Preparation of DNA for NCP assembly was performed as described[58]. Briefly, the plasmid described above was grown, isolated, and subjected to EcoRV digestion followed by separation of plasmid and insert by anion chromatography using Source 15Q resin (GE Healthcare). With the purified DNA, CENP-A NCPs were assembled and uniquely positioned using gradual salt dialysis followed by thermal shifting for 2 h at 55 °C (refs 60,61). Formation of complexes with CENP-C[CD] was performed as described[11], in which 2.2 moles of CENP-C[CD] were added per mole of CENP-A NCPs. To form the complex with CENP-N[NT], 4 moles of recombinant CENP-N[NT]-His were added per mole of CENP-A NCPs. To form the complex with both CENP-N[NT] and CENP-C[CD], 4 moles of CENP-N[NT]-His and 2.2 moles of CENP-C[CD] were added per mole of CENP-A NCPs. Complexes were analysed by 5% native PAGE, stained with ethidium bromide to visualize DNA and Coomassie Brilliant Blue to visualize protein components. Following formation of complexes (or NCPs, in the case of the nucleosome-alone sample), samples were purified by preparative electrophoresis (Prep Cell, BioRad) using a 5% native gel to isolate the relevant complex from other species, such as free DNA[60].

**Binding assays.** Recombinant human H2B K120C was purified as described for wild-type H2B (ref. 11) from inclusion bodies. Lyophilized protein was dissolved in unfolding buffer (7 M urea, 10 mM Tris-HCl pH 7.5 at 20 °C, 0.4 mM TCEP) for 1 hr at RT and a 15-fold molar excess of sulfo-Cy5-maleimide (Lumiprobe) was dissolved in dimethylsulphoxide and added dropwise to the protein. The reaction proceeded overnight shielded from light and was quenched with 5 mM sodium 2-sulfanylethanesulfonate (MESNA) and run over a PD-10 column (GE Healthcare) to separate out free dye. Labelled H2B was then mixed with equimolar amounts of H2A for dimer reconstitution and purification using established methods[15,60], but using SDS–PAGE gels to determine concentrations of H2A and labelled-H2B monomers for refolding. Three independent assays were performed for calculating apparent $K_d$ values for CENP-C WT and mutant proteins using CENP-A nucleosomes with labelled Cy5-H2B prepared on 147 bp DNA by gradient dialysis. Briefly, 200 nM of nucleosomes were incubated with increasing concentration of CENP-C WT or CENP-C mutants in buffer (20 mM Tris-Cl pH 7.5, 1 mM EDTA and 1 mM DTT) and incubated on ice for 1 hr before separating by 5% native PAGE. After electrophoresis, gels were analysed in a Typhoon 9200 imager (GE Healthcare), and the percentage of unbound vs. unbound nucleosomes were quantified using ImageJ. The apparent $K_d$ values were calculated from the binding curve fitted from three independent experiments.

**HXMS.** Deuterium on-exchange was carried out by adding 5 μl of each sample (containing ∼4 μg of NCPs or the indicated complex) to 15 μl of deuterium on-exchange buffer (10 mM Tris, pD 7.5, 0.5 mM EDTA, in D₂O) so that the final D₂O content was 75%. Reactions were quenched at the indicated timepoints by withdrawing 20 μl of the reaction volume, mixing in 30 μl ice-cold quench buffer (2.5 M GdHCl, 0.8% formic acid, 10% glycerol) and rapidly freezing in liquid nitrogen before proteolysis and liquid chromatography–mass spectrometry steps. HX samples were individually melted at 0 °C then injected (50 μl) and pumped through an immobilized pepsin (Sigma) column at initial flow rate of 50 μl min⁻¹ for 2 min followed by 150 μl min⁻¹ for another 2 min. Pepsin was immobilized by coupling to Poros 20 AL support (Applied Biosystems) and packed into column housings of 2 mm × 2 cm (64 μl) (Upchurch). Protease-generated fragments were collected onto a TARGA C8 5 μm Piccolo high-performance liquid chromatography column (1.0 × 5.0 mm, Higgins Analytical) and eluted through an analytical C18 liquid chromatography column (0.3 × 75 mm, Agilent) by a linear 12–55% buffer B gradient at 6 μl min⁻¹ (Buffer A: 0.1% formic acid; Buffer B: 0.1% formic acid, 99.9% acetonitrile). The effluent was electrosprayed into the mass spectrometer (LTQ Orbitrap XL, Thermo Fisher Scientific).

**HXMS data analysis.** The SEQUEST (Bioworks) software program was used to identify the likely sequence of parent peptides using non-deuterated samples via tandem MS. MATLAB-based MS data analysis tool, ExMS, was used for data

processing[62]. For all peptides found by SEQUEST, ExMS first analyses the non-deuterated sample to identify the peptide envelope centroid values as well as the chromatographic elution time ranges of each parental non-deuterated peptide. ExMS then uses the information from the non-deuterated analyses to identify deuterated peptides in each sample of the HXMS timecourse. Each individual deuterated peptide is corrected for loss of deuterium label during HXMS data collection (that is, back exchange after quench) by normalizing to the maximal deuteration level of that peptide, which we measure in a 'fully deuterated' (FD) reference sample. The FD sample are prepared in 75% deuterium just as is done in the on-exchange experiment, but under acidic denaturing conditions (0.5% formic acid), and incubated overnight, so that each amide proton undergoes full exchange. The extent of back-exchange is calculated by comparing the extent of full deuteration as measured in the FD sample to the theoretical maximal deuteration (that is, if no back-exchange occurs), which takes into account the 75% deuterium content of the samples. The median extent of back-exchange in our datasets is ∼12% (Supplementary Fig. 3e), which is within the range for the lowest amount of deuterium loss ever reported for bottom-up HXMS (10% ± 5%; ref. 63). For comparing two different HXMS datasets, we can plot the per cent difference of each peptide, which is calculated by subtracting the percent deuteration of one sample from that of another, and plotted according to the colour legend in 10% increments (as in Figs 2d, 4d, 5a and 6e). We can also calculate the number of deuterons within each peptide that are exchanged at each timepoint, and plotted as in Figs 4e,f and 5b–e, and Supplementary Figs 3f–j, 4a–i, 5f–i and 7. These plots include data from three separate exchange reactions, with each data point shown as mean ± s.d.

**Hydroxyl radical footprinting.** CENP-A nucleosomes assembled with HEX-labelled 147 bp α-satellite DNA[20] were reconstituted and then purified using a sucrose gradient. An amount of 4 μg of HEX-labelled CENP-A nucleosomes alone or complexed to CENP-N[NT] were used in each reaction. The hydroxyl radical cleavage reaction was initiated by addition of 5 μl of 40 mM FeAmSO₄/80 mM EDTA, 2 M ascorbate and 2.4% H₂O₂ to a 30 μl reaction mixture. Each reaction was carried out for 5 min at room temperature, and terminated with 200 μl of stop solution (0.1% SDS, 25 mM EDTA, 1% glycerol and 100 mM Tris, pH 7.4). Further phenol/chloroform extraction and ethanol precipitation was carried out to extract DNA fragments. Samples were separated by denaturing PAGE (10% polyacrylamide, 7 M urea, 88 mM Tris-borate and 2 mM EDTA, pH 8.3)[20]. Gels were imaged on a Typhoon 9200 imager (GE Healthcare). Band intensities were quantified from ImageJ from three independent experiments.

**Sucrose gradient sedimentation.** A total of 100 μg of CENP-A nucleosomes or the CCNC were subjected to 5–30% sucrose gradient centrifugation at 165,000g in a SW60 rotor (Beckman Coulter) for 13 h at 4 °C. The samples were fractionated from top to bottom, and each fraction was analysed for absorbance at 260 nm (for nucleosome and complex) or 280 nm (for CENP-C and CENP-N proteins alone).

**Data availability.** The authors declare that the data supporting the findings of this study are available within the paper and its Supplementary Information files, or are available from the authors upon reasonable request.

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

## Acknowledgements

We thank our UPenn colleagues M. Gerace, S. Falk and L. Mayne for assistance with experiments and M. Lampson for comments on the manuscript. We thank A. Straight (Stanford), K. Luger (Colorado) and F. Zheng (Addgene #42230) for plasmids. This work was supported by NIH research grants (GM082989, B.E.B.; GM108718, I.M.C; GM074150, D.W.C.), an NIH predoctoral fellowship (CA186430, L.Y.G.), and a post-doctoral fellowship from ACS (N.S.). L.Y.G. acknowledges support from the UPenn Structural Biology Training Grant (GM008275). G.A.L acknowledges support from UPenn Cell and Molecular Biology Training Grant (GM007229).

## Author contributions

L.Y.G. and B.E.B. conceived the study. L.Y.G., P.K.A., L.Z., K.L.M. and N.S. performed experiments. J.M.D.-M. provided key analytical tools. D.F., G.A.L., R.M.J. and D.W.C. provided key reagents. L.Y.G. and B.E.B. wrote the paper. L.Y.G., K.L.M., N.S., D.F., D.W.C., I.M.C. and B.E.B. edited the paper.

## Additional information

**Competing interests:** The authors declare no competing financial interests.

