## [Peer Review File · Nature Communications]

Reviewers' comments:

Reviewer #1 (Remarks to the Author):

The authors substantially changed manuscript adding new data and performing statistical analysis of experimental results. My only criticism with the revision is with the authors' interpretation of the results in terms of structural transitions of nucleosomes and their dynamics upon the interaction with CENP-C which is primarily described in section "The arginine anchor of CENP-CCD is required for the CENP-A nucleosome structural transition and stabilizing centromeric chromatin" (pp. 6-8). The data in the paper clearly show the interaction of CENP-C with the nucleosomes and HXMX approach allows the authors to identify the contact areas between histones and CENP-C, but these data do not provide structural details of nucleosomes and their structural transitions upon interaction with CENP-C. More questionable are statements on "altered dynamics" of nucleosomes (p. 8). Along with published data, the data in the paper can be discussed in terms of structure and dynamics, but it should be carefully explained and justified and placed in the discussion part of the paper.

Reviewer #2 (Remarks to the Author):

Trying to address my question about the stability of the complexes, I disagree with the author's statement on page 12 top where they say that the fact they didn't see any EX1 kinetics in the MS spectra is an indication that the complex is stable and there are no free unbound species in their mixture.

I think this statement is completely speculative as EX1 kinetics doesn't necessarily occur only in these cases (there is quite a lot of HX MS literature) that indicate that EX1 kinetics can occur regardless of the bound or unbound state of a protein; it has to do with the protein dynamics and intrinsic exchange and has nothing to do with the fact that a complex is formed or not. HX goes to completion with each opening event of the protein, which is described as EX1 kinetics.

Page 22; fully deuterated sample: as I mentioned before the fact that there is only 75% D2O final content is worrisome because in the presence of back exchange the interpretation of the results might be erroneous. The author's response that all their previous work was performed in the same way (75% D2O final) is worrisome. They claim that they got only 12% back exchange? I think it's very difficult to make a fully deuterated sample, and even at "equilibrium" because it's an acid-base catalysis there is still D loss; therefore I would like to see how the authors did this calculation that based on a 75% D2O they have only 12% back exchange. They also say that the median deuteration was 88% in the case of a fully deuterated sample; I think this can be very deceiving because they look at deuteration of individual peptides; of course based on the amino acid sequence some of the peptides would retain more and some less deuterium; therefore the fact that they say that their back exchange was only 12% is really concerning. In the same paragraph they state that "the lowest amount of deuterium loss ever reported is (10% ±5%)"; this statement is not true; there are HX MS publications by Borchers et al. that report a 2% back exchange. I would recommend that either the authors remove that statement or at least cite the relevant literature.

I still have a big problem with the fact that now Figure 6 (e and f) shows the same or very similar data that was presented in Figure 2 a and b of the Science 2015 Falk et al. paper. The only small difference is that they have more peptides now; but the conclusions are the same. However; on Figure 6 e, when they show the peptides that are protected in CENP-A construct, $\alpha 3$ region; why aren't the two grey peptides in the black box also blue? They correspond to the same region as the 4 peptides shown in blue below.

Same comment for H4 $\alpha 3\beta$ region, the peptides in the dotted black box, there is one grey peptide in the overlapping series of 4 blue peptides; why is that??? Same problem in H2A; there are some overlapping peptides in $\alpha 2$; $\alpha 3$ and β regions that show protection and some don't; this is very

unusual and I think it's incorrect. The overlapping peptides should behave similar. I would like to see all the deuterium incorporation graphs for all the peptides displayed in Figure 6e.

I disagree with the authors statement that if they didn't see any binding with the W530A mutant they shouldn't perform an HX MS experiment on that construct bound to CENP-A. That would be a very nice negative control and I recommend that they perform that experiment.

In the new Supplemental Figure 6 (a and b) I would like to see all the deuterium graphs in order to better visualize exactly which peptides get slightly protected; it's difficult to assess the data in the way its displayed here as some of the peptides are really short.

Also, I do not completely agree with the authors explanation of why they didn't perform an HX MS comparison between CENP-NNT full and CENP-N1-205. What happens when the molecule is truncated? I think the folding of the full length protein might be different and maybe the data would show additional new insights into the binding to CENP-A nucleosomes.

Figure 2 d; in the H4 β region that shows 3 protected peptides; why aren't the three peptides above also protected? They cover the same region and are overlapping peptides! Same comment for H2A α 2- α 3 region, where only two peptides are shown to be protected in blue, however the other peptides that are overlapping don't show any changes; this cannot be correct! I would like to see all the graphs of all overlapping peptides in the supplement.

Similar comment for Figure 5a; in the region between amino acids 200-220 there are two peptides that are protected (in blue) but other overlapping peptides covering that same exact area are not. Why is that? Also, I think that where authors indicate the graph showed in Figure 5e is not correct; in the peptide map where "e" is indicated there is no peptide showing protection. The display is incorrect.

Reviewer #3 (Remarks to the Author):

The authors have done a thorough job of addressing my concerns and the additional data have strengthened the core conclusions of the work. I recommend publication after correcting the following minor omission.

I had made the suggestion below for improving clarity and ensuring that it was clear to the reader that the parts of the conclusion being stated were known from previous work:

The statement on page 12 "CENP-CCD and CENP-NNT, therefore, defy expectations in that the unfolded one (CENP-CCD) generates substantial structural changes in the NCP whereas the folded one (CENP-NNT) changes core histone dynamics only very locally at the point of contact with CENP-A" could be changed to something like: "Therefore, the combination of previous work and the work presented here shows that CENP-CCD and CENP-NNT, defy expectations in that the unfolded one (CENP-CCD) generates substantial structural changes in the NCP whereas the folded one (CENP-NNT) changes core histone dynamics only very locally at the point of contact with CENP-A"

The authors mention that they incorporated this suggestion but the revised manuscript still seems to have the original sentence on page 15.

Reviewers' comments and **our responses**:

Reviewer #1 (Remarks to the Author):

The authors substantially changed manuscript adding new data and performing statistical analysis of experimental results. My only criticism with the revision is with the authors' interpretation of the results in terms of structural transitions of nucleosomes and their dynamics upon the interaction with CENP-C which is primarily described in section "The arginine anchor of CENP-CCD is required for the CENP-A nucleosome structural transition and stabilizing centromeric chromatin" (pp. 6-8). The data in the paper clearly show the interaction of CENP-C with the nucleosomes and HXMX approach allows the authors to identify the contact areas between histones and CENP-C, but these data do not provide structural details of nucleosomes and their structural transitions upon interaction with CENP-C. More questionable are statements on "altered dynamics" of nucleosomes (p. 8). Along with published data, the data in the paper can be discussed in terms of structure and dynamics, but it should be carefully explained and justified and placed in the discussion part of the paper.

We thank the reviewer for acknowledging the new data and analysis we performed and are happy that he/she only has criticism of this aspect of the writing. We use the HX protection of the internal β -sheet between H4 and H2A as a read-out for the shape change that CENP-C imparts. Thus, we disagree with the assertion of the reviewer that HXMS only reads the contact area with CENP-C^{CD}. The data itself from the WT complex (studied in Falk et al., 2015, *Science*) show this quite clearly: there is HX protection that maps unambiguously to the buried center of the nucleosome to a region encapsulating a β -sheet that forms between H2A and H4, coinciding with the CENP-A nucleosome structural transition. We agree, though, that this needed to be more clearly laid out in introducing the rationale for the new experiments, and, later, when we discuss their implications. To first address the concern of the reviewer, we adjusted the motivation of the experiment in the sub-section of the Results section referred to by the Reviewer, so that our previous evidence for observing internal changes within the buried core of the histone octamer was clearly explained as our motivation to probe this aspect of the nucleosome structural transition with HXMS [see changes on pg. 7]. We also agree with the reviewer's view that the implications of our new HXMS findings in this section of the paper is best spelled out and connected to the previous findings in the published literature in the 'Discussion' section of the paper. We have removed or revised many sentences in the 'Results' section on pgs 8-9. We also appreciate that "altered dynamics" could potentially refer to backbone dynamics on the histones (HXMS experiments) or the turnover of CENP-A nucleosomes at centromeres (SNAP experiments). So we have clarified which aspect we're discussing in the text in two key places on pg 8.

Reviewer #2 (Remarks to the Author):

Trying to address my question about the stability of the complexes, I disagree with the author's statement on page 12 top where they say that the fact they didn't see any EX1 kinetics in the MS spectra is an indication that the complex is stable and there are no free unbound species in their mixture. I think this statement is completely speculative as EX1 kinetics doesn't necessarily occur only in these cases (there is quite a lot of HX MS literature) that indicate that EX1 kinetics can occur regardless of the bound or unbound state of a protein; it has to do with the protein dynamics and intrinsic exchange and has nothing to do with the fact that a complex is formed or not. HX goes to completion with each opening event of the protein, which is described as EX1 kinetics.

There is a misunderstanding here: we never mentioned "EX1 kinetics" in the text of the paper (on pg. 11-12 or anywhere else in the manuscript) and we stand by our explanation in the text that the lack of a bimodal population is evidence from within our HXMS experiment that the complex is stable for at least 100,000s. And to hopefully remove any possibility of future confusion, we changed the sentence as follows for the relevant sentence on pg. 11: "without any evidence of bimodal peaks or any other fast exchanging species that could have corresponded to a substantially populated unbound, unprotected state". We are well aware of sources of true EX1 behavior in monomeric proteins and the possible sources of EX1-like kinetics in multi-protein complexes, like the one under investigation. The bottom line is we don't see them and we don't even mention EX1. In our Figure 6d, we show a representative peptide from our CCNC complex that is remarkably protected from exchange across all timepoints, indicating that the complex does not "open" or transiently disassemble to allow for HX to go "to completion". Furthermore, in addition to our HXMS data, we have additional lines of evidence indicating that our complex is stable, including data showing that it co-elutes through a preparatory native gel (Supplementary Fig 5c) and a sucrose gradient (Supplementary Fig 5d,e).

Page 22; fully deuterated sample: as I mentioned before the fact that there is only 75% D2O final content is worrisome because in the presence of back exchange the interpretation of the results might be erroneous. The author's response that all their previous work was performed in the same way (75% D2O final) is worrisome. They claim that they got only 12% back exchange? I think it's very difficult to make a fully deuterated sample, and even at "equilibrium" because it's an acid-base catalysis there is still D loss; therefore I would like to see how the authors did this calculation that based on a 75% D2O they have only 12 % back exchange. They also say that the median deuteration was 88% in the case of a fully deuterated sample; I think this can be very deceiving because they look at deuteration of individual peptides; of course based on the amino acid sequence some of the peptides would retain more and some less deuterium; therefore the

fact that they say that their back exchange was only 12% is really concerning. In the same paragraph they state that “the lowest amount of deuterium loss ever reported is (10% ±5%)”; this statement is not true; there are HX MS publications by Borchers et al. that report a 2% back exchange. I would recommend that either the authors remove that statement or at least cite the relevant literature.

Again, there appears to be a fundamental misunderstanding. Preparing samples in 75% final deuterium content is the standard approach in the field used by pioneers in the field, such as David Smith, Virgil Woods, Walter Englander, and others (Zhang and Smith, *Protein Sci* 1993; Tobler et al., *Protein Sci* 2002; Del Mar et al., *PNAS* 2005; Hsu et al., *JBC* 2008; Burke et al., *J Am Chem Soc* 2009; Hailey et al., *JBC* 2009; Walters et al., *J. Am. Soc. Mass Spectrom* 2012; *Methods in Enzymology*, Vol 557, ed. A. Shukla 2015, and many more...).

As we detailed in the Methods, already revised in the previous version in response to the reviewer’s earlier misunderstanding, each individual deuterated peptide is corrected for back-exchange by normalizing to an experimental “fully deuterated” (FD) reference sample. The FD sample is prepared in 75% final deuterium content, just as is done in the on-exchange experiment. Preparing samples as 75% final deuterium content means that 25% of the volume simply comes from the volume of the nucleosome complex in its native buffer. To achieve 100% final deuterium content, you would first need to lyophilize the complex (or use some other such method) and reconstitute in 100% D₂O, which is not realistic for the CCNC. Moreover, it's unnecessary to achieve 100% deuteration since back-exchange is corrected for on a per peptide basis. The bottom line is that we know that any individual amide proton exchange event has a 75% chance of exchanging with a deuterium and a 25% chance of exchanging with a proton. This doesn’t negatively affect our experiments. We account for the usage of 75% D₂O for the on-exchange reaction in every step of the experiment, and in every calculation. This is very easy to do (see example calculations on the next page). Perhaps this is difficult for some experiments to measure each position by an ETD approach...but that isn’t what we’re doing because it has never been shown to work at the scale we require to interrogate the CCNC.

We detailed in the Methods that our extent of back-exchange is calculated by comparing the extent of full deuteration *as measured in the FD sample* to the *theoretical max deuteration (i.e., if no back-exchange occurs)*. Since this point caused confusion for the reviewer, we have now clarified in the text that in our calculation, the “theoretical max deuteration” already takes into account the 75% deuterium content of the samples. (see pg. 22). The “theoretical max deuteration” is calculated as: the number of amino acids in the peptide (except for proline, which does not contain an amide proton) subtracted by 2, since exchange of the first two backbone amide protons cannot be measured (Bai et al., *Proteins*

1993) multiplied by the % deuterium in the sample (which, in our case, is 75%).

Below is a step-by-step example of how we would manually calculate the extent of deuteration for a sample peptide (CENP-A a.a. 74-84 charge state 1; sequence is ICVKFTRGVDF) in a representative fully-deuterated (FD) sample:

The peptide has 11 amino acids (and no prolines),
therefore its “theoretical max deuteration” = $(11-2)*0.75$

The “actual deuteration” is calculated as [(Centroid m/z of the peptide in the fully deuterated sample) – (Centroid m/z of the peptide in the non-deuterated sample)]*(charge state of peptide).

The centroid m/z values of the peptides are calculated by the ExMS software (Kan et al., 2011) from each of our mass spec runs:

The centroid m/z of the peptide in the non-deuterated sample is 1285.429
The centroid m/z of the peptide in the fully deuterated sample is 1291.523

Therefore, the “actual deuteration” is $(1291.523-1285.429)*1 = 6.094$

Extent of deuteration = $6.094/((11-2)*0.75) = 90.3\%$

Back-exchange of this peptide = $100 - 90.3 = 9.7\%$

The above calculations are performed for *every* peptide in the fully deuterated (FD) sample, using EXMS in Matlab to automate the analysis. Supplementary Fig 3e, as we had stated in the figure legends, shows a cumulative distribution curve of all peptides in a representative fully deuterated (FD) sample. The graph clearly shows that the *median* extent of deuteration is 88%, which by definition means that some peptides retained more and some less deuterium (as this reviewer pointed out). We had elected to show this entire cumulative distribution (instead of simply reporting a statistic, e.g., mean±SD) to precisely show the complete extent of deuteration across our entire peptide pool. We have also now improved the labeling of Supplementary Fig 3e to clarify how we do these calculations.

Correction for back-exchange is performed on a per peptide basis, per the following formula:

$$\% \text{ Deuteration} = \frac{(\text{centroid m/z timepoint}) - (\text{centroid m/z ND})}{(\text{centroid m/z FD}) - (\text{centroid m/z ND})} * 100\%$$

Below is an example calculation, using our example peptide from above. At the 10²s timepoint in one dataset, the centroid m/z of this peptide is 1288.432. Therefore:

$$\% \text{ Deuteration at the } 10^2 \text{ s timepoint} = \frac{(1288.432) - (1285.429)}{(1291.523) - (1285.429)} * 100\% = 49.3\%$$

As you can see, it is very arithmetically simple to use the fully-deuterated (FD) sample to account for back-exchange in our dataset.

The Borchers publications cited by this reviewer (such as Pan and Borchers, *Proteomics* 2013) refer to a related— yet very different—method of HXMS. In our method, which is the traditional “bottom-up” approach, the deuterated proteins undergo in-line enzymatic digestion performed at conditions carefully optimized to minimize back-exchange prior to LC-MS separation (per Walters et al., *J. Am. Soc. Mass Spectrom* 2012). Borchers et al uses a “top-down” mass spec approach that fragments intact proteins by electron capture dissociation (ECD). Because this method uses direct whole molecule sample injection and bypasses the enzymatic digestion step, this reduces the amount of back-exchange. However, we think most in the field would agree with us that currently the bottom-up approach is more successful for analyzing large proteins and complexes, due to relative difficulty in analyzing large intact protein complexes top-down. We thank this reviewer for his/her suggestion to be more precise in our language and specify that we are comparing our back-exchange to that of bottom up (and not top-down) approaches. We have added the words “bottom up” to that sentence in the text:

“...the back exchange in our datasets is ~12%, which is within the range for the lowest amount of deuterium loss ever reported for bottom-up HXMS.” (see pg. 22)

A recent publication by Borchers directly compares bottom-up vs. top-down HXMS for the same proteins (Pan, Zhang, Borchers, *Biochimica et Biophysica Acta* 2016). They report that their back-exchange level was “10-50%” for the bottom-up approach, and lower (2%) for the top-down approach. Therefore, our back-exchange level is well within their bottom-up range. They also stated that “top-down and bottom-up data were consistent with each other and provided complementary information.”

We believe that our additions as described above have thoroughly addressed this reviewer’s concerns about how we addressed back-exchange in this study.

I still have a big problem with the fact that now Figure 6 (e and f) shows the same or very similar data that was in presented in Figure 2 a and b of the Science 2015 Falk et al. paper. The only small difference is that they have more peptides now; but the conclusions are the same.

Here, there is a really deep misunderstanding. As we had clarified in this previous rebuttal, Figure 6 shows HXMS data on the complex of CENP-A nucleosomes bound to CENP-C and CENP-N simultaneously, *which is a complex on which there exists no published*

structural/dynamic data. We remain mystified why the reviewer thinks the CCNC work has already been published. The Falk et al paper referred to by the reviewer had absolutely no data about CENP-N whatsoever! Expectedly, some features, but not others, are common between the CCNC and the complex described in Falk et al. And, of course, we display our data in the same format precisely to help our readers contextualize our findings with the existing literature in a straightforward and transparent way. But no part of Figure 6e,f has ever been reported.

However; on Figure 6e, when they show the peptides that are protected in CENP-A construct, $\alpha 3$ region; why aren't the two grey peptides in the black box also blue? They correspond to the same region as the 4 peptides shown in blue below. Same comment for H4 $\alpha 3\beta$ region, the peptides in the dotted black box, there is one grey peptide in the overlapping series of 4 blue peptides; why is that??? Same problem in H2A; there are some overlapping peptides in $\alpha 2$; $\alpha 3$ and β regions that show protection and some don't; this is very unusual and I think it's incorrect. The overlapping peptides should behave similar. I would like to see all the deuterium incorporation graphs for all the peptides displayed in Figure 6e.

Once again there is a misunderstanding here, and if our speculation is correct on the source of the confusion, then there is a simple explanation. We speculate that this reviewer's confusion stems from not realizing that Fig 6e is a 'difference plot'. As we detailed in the Methods, these plots display the "percent difference of each peptide, which is calculated by subtracting the percent deuteration of one sample from that of another, and plotted according to the color legend in 10% increments." They are in a similar format to what has been used us and others for many years in many different publications: see Wang et al., *JBC* 2010; Panchenko et al., *PNAS* 2011; Landgraf et al., *Structure* 2013; DeNizio et al., *Nucleic Acid Res* 2014; Dawicki-McKenna et al *Mol Cell* 2015; Falk et al., *Science* 2016; Fanning et al., *eLife* 2016, and many more.

In any plots like this, there are inevitably some partially overlapping peptides that are on the cusp and on either side of the 10% cut-off. Also, this means that for similar changes in the number of deuterons, the percent difference will be smaller for *longer* peptides compared to *shorter* peptides when the a.a. residue(s) protected from exchange fall entirely/mostly in the region spanned by the shorter one. For example, in the boxed CENP-A $\alpha 3$ region, these are the 4 peptides that appear blue. They are 10-11 residues long, and their increase in protection upon binding to CENP-C and CENP-N is >10%. The two "grey peptides" are two longer peptides (26 residues) that span this region, but their percent differences are <10%. Therefore, they appear as grey in our difference plot, as per the color legend of the figure. And for the H4 and H2A regions mentioned by this reviewer, the percent differences are right on the cusp of the 10% cut-off, so an occasional peptide may fall right beneath this cut-off.

In light of all of this, the reviewer's comments are irrelevant to our claims in the paper. By any reasonable comparison to published HXMS experiments, we have excellent peptide coverage of a large macromolecular complex, the CCNC. We don't draw our conclusions from any single peptide that behaves differently than its overlapping peptides. Instead, we look at the trends across multiple overlapping peptides, as well as across multiple datasets, and expand on this with statistical analyses of representative peptides in every region of relevance/interest (for just some examples, see Supplementary Fig 5f-i).

I disagree with the authors statement that if they didn't see any binding with the W530A mutant they shouldn't perform an HX MS experiment on that construct bound to CENP-A. That would be a very nice negative control and I recommend that they perform that experiment.

We disagree with the perceived niceness of such an extension of our paper. What would this accomplish? To see how a mutant protein that doesn't detectably bind a nucleosome affects a nucleosome? We do not view this as an important experiment, and we do not think it would add anything of value to the paper. Certainly nothing that should delay the publication of our study. This mutant was used to address another reviewer's (Reviewer #3's) wish for us to have this mutant to compare to the key mutant in our paper (R522A). We did this along with the many other impactful extensions made prior to the previous round of peer review. The reviewer (Reviewer #3) is satisfied.

In the new Supplemental Figure 6 (a and b) I would like to see all the deuterium graphs in order to better visualize exactly which peptides get slightly protected; it's difficult to assess the data in the way its displayed here as some of the peptides are really short.

Figure S6 is a very minor point of the paper. It is referred to in a single sentence: "CENP-C^{CD}, itself, undergoes rapid HX, consistent with CENP-C^{CD} existing as a primarily linear polypeptide lacking defined secondary structure, although there is reduced HX at the earliest timepoints within ~a.a. 515-537 when bound to CENP-A nucleosomes (Supplementary Fig. 6)." The paper would be fine even without any of this and the fact that CENP-C^{CD} lacks its own secondary structure. The main conclusions of the paper would be 100% the same whether or not we did this experiment at all. The protection we report is minor (as was also shown by the peptide examples we had in Fig. S5 of the original version the reviewer critiqued at another Nature journal), which is pasted below for reference. In sum, we are happy with the new data display for S6—generated in response to the initial set of reviews—and we think it made the paper better (we're happier with S6 than the previous figure for this experiment) while keeping it clear to the reader that this is not a major point of the paper.

Legend for this earlier Supplementary Figure that has been replaced by the current S6. CENP-C^{CD} lacks detectable secondary structure and binds the histone surface of CENP-A nucleosomes with residues ~515-537. (a) Diagram of CENP-C^{CD} (a.a. 426-537), indicating the region previously reported to bind the protein surface of the nucleosome (Kato et al., 2013). (b-j) Peptides encompassing various regions within CENP-C^{CD}, showing CENP-C^{CD} alone and CENP-C^{CD} bound to CENP-A nucleosomes. CENP-C^{CD} lacks any secondary structure (Kato et al., 2013), therefore when it is not bound to CENP-A nucleosomes, all regions fully exchange even at 10¹ s at 4°C (the reduced temperature slows the chemical exchange rate allowing HX changes on regions of proteins not protected by stable secondary structure). Nonetheless, the a.a. ~515-537 region exhibit protection from HX only at the earliest timepoints, which corresponds to the region reported to bind to the protein surface of the nucleosome (Kato et al., 2013). We note that in panel e, the peptide was not detected in the CENP-C^{CD}-alone sample, but we included it in the figure nonetheless because it spans a region of CENP-C^{CD} that does not display any HX protection, even in the presence of the CENP-A nucleosome. The maximum number of deuterons possible to measure by HXMS for each peptide is shown by the dotted line.

Also, I do not completely agree with the authors explanation of why they didn't perform an HX MS comparison between CENP-NNT full and CENP-N1-205. What happens when the molecule is truncated? I think the folding of the full length protein might be different and maybe the data would show additional new insights into the binding to CENP-A nucleosomes.

This reviewer seems to misunderstand our experiment. What he/she refers to as “CENPNNT full” is CENP-N 1-240, which is the N-terminal nucleosome-binding portion of CENP-N. The only difference between that and CENP-N 1-205 is the region between residues 206-240, which *does not appear to engage the CENP-A nucleosome* (which we show in Supplementary Figures 4 and 7). Therefore, it is highly unlikely that additional data from a molecule that is missing a region that never binds the CENP-A nucleosome in the first place would provide additional insight into the binding to CENP-A nucleosomes. Therefore, we think that anything of this sort is outside of the scope or interest of our team on this paper.

Figure 2 d; in the H4 β region that shows 3 protected peptides; why aren't the three peptides above also protected? They cover the same region and are overlapping peptides! Same comment for H2A $\alpha 2$ - $\alpha 3$ region, where only two peptides are shown to be protected in blue, however the other peptides that are overlapping don't show any changes; this cannot be correct! I would like to see all the graphs of all overlapping peptides in the supplement. Similar comment for Figure 5a; in the region between amino acids 200-220 there are two peptides that are protected (in blue) but other overlapping peptides covering that same exact area are not. Why is that?

Here is another misunderstanding. Again, there is a simple explanation: the same one to one of this reviewer's other comments (the one on Fig. 6e/f, above). Figure 2d and Figure 5a are all difference plots, showing the “percent difference of each peptide, which is calculated by subtracting the percent deuteration of one sample from that of another, and plotted according to the color legend in 10% increments.” Therefore, there are some peptides on the cusp of the 10% cut-off. Also, the percent difference will be smaller for *longer* peptides compared to *shorter* peptides. In the H4 β region, the three peptides above the 3 blue peptides are longer and fell just below the 10% cut-off. As above, this reviewer's comments are basically irrelevant to our claims of the paper, since we don't draw our conclusions from any single peptide that behaves differently than its overlapping peptides. Instead, we look at the trends across multiple overlapping peptides, as well as across multiple datasets, and had expanded on this with statistical analyses of representative peptides in every region of relevance/interest (see Fig 2e-i, 4e-f, 5b-e, Supplementary Fig 3f-i, 4a-i, 5f-i, 7)

Also, I think that where authors indicate the graph showed in Figure 5e is not correct; in the peptide map where "e" is indicated there is no peptide showing protection. The display is incorrect.

Another misunderstanding. The display is NOT incorrect. The peptide map in Fig 5a is from the 10^4 timepoint (which we clearly stated in the figure legends). If you look at Fig 5e, you can clearly see that at the 10^4 timepoint, there is no difference in protection. Nonetheless, to clarify for readers who do not read figure legends, we have now added “ 10^4 s” to the figure itself in Fig 5a. We also did the same for all other difference plots in the paper: by labeling the figure itself with the timepoint (instead of just having that in the figure legends).

Reviewer #3 (Remarks to the Author):

The authors have done a thorough job of addressing my concerns and the additional data have strengthened the core conclusions of the work. I recommend publication after correcting the following minor omission.

I had made the suggestion below for improving clarity and ensuring that it was clear to the reader that the parts of the conclusion being stated were known from previous work:

The statement on page 12 “CENP-CCD and CENP-NNT, therefore, defy expectations in that the unfolded one (CENP-CCD) generates substantial structural changes in the NCP whereas the folded one (CENP-NNT) changes core histone dynamics only very locally at the point of contact with CENP-A” could be changed to something like: “Therefore, the combination of previous work and the work presented here shows that CENP-CCD and CENP-NNT, defy expectations in that the unfolded one (CENP-CCD) generates substantial structural changes in the NCP whereas the folded one (CENP-NNT) changes core histone dynamics only very locally at the point of contact with CENP-A”

The authors mention that they incorporated this suggestion but the revised manuscript still seems to have the original sentence on page 15.

We thank this reviewer for recommending the publication of this manuscript, and apologize for this unintended oversight in our previous revision. We have now modified the statement on page 15 per this reviewer’s suggestion, and also cited the specific publications.

REVIEWERS' COMMENTS:

Reviewer #1 (Remarks to the Author):

The authors responded to my critiques.

Reviewer #2 (Remarks to the Author):

The authors addressed most of my concerns and finally explained well what they did. The "misunderstanding" that the authors were talking about was generated by the fact that some of the findings in their manuscript were poorly written/ explained before. As it stands now I think that this manuscript can be published in your journal without any further modifications.

Reviewer #3 (Remarks to the Author):

Recommend for publication.